# Cryo-EM structure of the lysosomal chloride-proton exchanger CLC-7 in complex with OSTM1

**Marina Schrecker, Julia Korobenko, Richard K Hite\***

Structural Biology Program, Memorial Sloan Kettering Cancer Center, New York, United States

**Abstract** The chloride-proton exchanger CLC-7 plays critical roles in lysosomal homeostasis and bone regeneration and its mutation can lead to osteopetrosis, lysosomal storage disease and neurological disorders. In lysosomes and the ruffled border of osteoclasts, CLC-7 requires a β-subunit, OSTM1, for stability and activity. Here, we present electron cryomicroscopy structures of CLC-7 in occluded states by itself and in complex with OSTM1, determined at resolutions up to 2.8 Å. In the complex, the luminal surface of CLC-7 is entirely covered by a dimer of the heavily glycosylated and disulfide-bonded OSTM1, which serves to protect CLC-7 from the degradative environment of the lysosomal lumen. OSTM1 binding does not induce large-scale rearrangements of CLC-7, but does have minor effects on the conformation of the ion-conduction pathway, potentially contributing to its regulatory role. These studies provide insights into the role of OSTM1 and serve as a foundation for understanding the mechanisms of CLC-7 regulation.

## Introduction

CLC-7 is a member of the CLC family of chloride (Cl$^-$) channels and chloride (Cl$^-$)/proton (H$^+$) transporters and is expressed in the lysosome and the resorption lacuna of osteoclasts (*Graves et al., 2008*; *Ishida et al., 2013*; *Kornak et al., 2001*; *Weinert et al., 2010*). In the membranes of these acidic compartments, CLC-7 uses the large pH gradient to catalyze the uptake of two Cl$^-$ ions for each H$^+$ released (*Graves et al., 2008*; *Leisle et al., 2011*; *Ludwig et al., 2013*). Dysfunction of CLC-7 is associated with dysregulation of ion and pH homoeostasis of the lysosome and the resorption lacuna (*Graves et al., 2008*; *Ishida et al., 2013*; *Kasper et al., 2005*; *Kornak et al., 2001*; *Lange et al., 2006*; *Undiagnosed Diseases Network et al., 2019*; *Steinberg et al., 2010*; *Weinert et al., 2010*). As both of these compartments rely on high proton concentrations to perform their physiological roles, disruption of CLC-7 function is associated with human diseases driven by impaired lysosomal and/or osteoclast function (*Kasper et al., 2005*; *Kornak et al., 2001*; *Lange et al., 2006*; *Undiagnosed Diseases Network et al., 2019*; *Pressey et al., 2010*). In particular, osteopetrosis, a disease characterized by dense and brittle bones, is the most common disease associated with CLC-7 mutation, with more than 50 distinct pathogenic mutations identified to date (*Chalhoub et al., 2003*; *Cleiren et al., 2001*; *Kasper et al., 2005*; *Kornak et al., 2001*; *Lange et al., 2006*; *Sartelet et al., 2014*; *Schulz et al., 2010*; *Weinert et al., 2010*).

Extensive structural and functional characterization of prokaryotic and eukaryotic CLC channels and transporters have established a framework for Cl$^-$/H$^+$ exchange and identified several key residues that participate in the transport cycle (*Accardi et al., 2004*; *Accardi and Miller, 2004*; *Accardi et al., 2005*; *Basilio et al., 2014*; *Chavan et al., 2020*; *Dutzler et al., 2002*; *Dutzler et al., 2003*; *Feng et al., 2010*; *Feng et al., 2012*; *Jayaram et al., 2008*; *Park et al., 2017*; *Park and MacKinnon, 2018*; *Picollo et al., 2012*; *Wang et al., 2019*; *Zdebik et al., 2008*). Within the Cl$^-$-conduction pathway, the gating glutamate (Glu$_{gate}$) that is conserved in CLC transporters is proposed to

**\*For correspondence:**
hiter@mskcc.org

**Competing interests:** The authors declare that no competing interests exist.

**eLife digest** Inside the cells of mammals, acidic compartments called lysosomes are responsible for breaking down large molecules and worn-out cells parts so their components can be used again. Similar to lysosomes, specialized cells called osteoclasts require an acidic environment to degrade tissues in the bone. Both osteoclasts and lysosomes rely on a two-component protein complex to help them digest molecules. Mutations in the genes for both proteins are directly linked to human diseases including neurodegeneration and osteopetrosis – a disease characterized by dense and brittle bones.

For the main protein in this complex, called CLC-7, to remain stable and perform its roles, it requires an accessory subunit known as OSTM1. CLC-7 is a transporter that funnels electrically charged particles into and out of the lysosome, which helps to maintain the environment inside the lysosome compartment. However, due to the tight partnership between CLC-7 and OTSM1, how they influence each other is poorly understood.

To determine the roles of CLC-7 and OSTM1, Schrecker et al. looked at the structure of the complex using a technique called single particle electron microscopy, which allows proteins to be visualized almost down to the individual atom. The analysis revealed that OSTM1 covers almost the entire inside surface of CLC-7, protecting it from the acidic environment inside the lysosome and contributing to its stability. When the two subunits are bound together, OSTM1 also slightly changes the structure of the pore formed by CLC-7, suggesting that OSTM1 may regulate CLC-7 activity.

Schrecker et al. have laid the foundation for understanding more about the activity and regulation of CLC-7 and OSTM1 in lysosomes and osteoclasts. The structures described also help explain previous findings, including why OSTM1 is important for the stability of CLC-7.

oscillate between at least four different conformations (*Chavan et al., 2020*; *Dutzler et al., 2002*; *Dutzler et al., 2003*; *Feng et al., 2010*). The movement and changes in the protonation state of $Glu_{gate}$ are coupled to the binding and release of $Cl^-$ ions in the highly conserved external and central binding sites (*Picollo et al., 2012*). Near the center of the transporter, the anion and $H^+$-conduction pathways diverge with the anion pathway passing through the internal binding site before reaching the cytosol, while the $H^+$-conduction pathway passes through a hydrophobic gap before reaching a conserved internal glutamate ($Glu_{in}$) (*Accardi and Miller, 2004*; *Accardi et al., 2005*; *Chavan et al., 2020*; *Leisle et al., 2020*; *Lim and Miller, 2009*; *Zdebik et al., 2008*). This conserved $Glu_{in}$ is dispensable for coupled transport and water molecules has been proposed to mediate $H^+$ transport through the hydrophobic gap (*Feng et al., 2010*; *Han et al., 2014*; *Wang and Voth, 2009*). Despite these extensive efforts, the precise mechanisms by which $Cl^-$ and $H^+$ transport are coupled remains poorly understood as are the mechanisms that underlie the gating of CLC transporters.

Unique among mammalian CLC transporters, CLC-7 requires a β-subunit, osteopetrosis-associated transmembrane protein 1 (OSTM1), for transport activity (*Lange et al., 2006*; *Leisle et al., 2011*). CLC-7 and OSTM1 co-localize in lysosomes and the ruffled border of osteoclasts (*Lange et al., 2006*; *Leisle et al., 2011*; *Schulz et al., 2010*). There, CLC-7 and OSTM1 stabilize the expression of one another and are both required for $Cl^-/H^+$ exchange (*Lange et al., 2006*; *Leisle et al., 2011*). OSTM1 is predicted to be a glycosylated, single-pass transmembrane protein and mutations in OSTM1, like mutations in CLC-7, can lead to osteopetrosis and neurodegeneration in humans and mice (*Chalhoub et al., 2003*; *Kasper et al., 2005*; *Kornak et al., 2001*; *Lange et al., 2006*; *Majumdar et al., 2011*; *Pressey et al., 2010*). However, the mechanisms by which OSTM1 and CLC-7 cooperate to enable proper ion transport remains an open question. To begin to understand the mechanisms of CLC-7 function and its unique requirement for OSTM1, we have determined electron cryomicroscopy (cryo-EM) structures of CLC-7 and of a CLC-7/OSTM1 complex.

## Results

### Structure of CLC-7

Following an evaluation of multiple CLC-7 orthologues, we decided to focus our structural studies on the chicken and human CLC-7 proteins based on their expression levels and their biochemical stabilities. Full-length chicken CLC-7 (ggCLC-7) and human CLC-7, which are 86.4% identical, were expressed in HEK293S GnTI- cells as mEGFP-fusions, purified to homogeneity in the detergent lauryl maltose neopentyl glycol (LMNG), cholesterol hemisuccinate (CHS), 150 mM KCl and 50 mM Tris-HCl pH 8.0, and analyzed by cryo-EM. Vitrified human CLC-7 transporters displayed a strongly preferred orientation that was confirmed by two-dimensional classification (*Figure 1—figure supplement 1*). Because of the very limited views of the transporter, we were not able to reconstruct a three-dimensional density map of human CLC-7. In contrast, two-dimensional classification of ggCLC-7 revealed a wide range of views and was suitable for three-dimensional structure determination (*Figure 1—figure supplement 2*). Three-dimensional classification of the imaged ggCLC-7 transporters identified a single class that displayed both well-ordered transmembrane and cytosolic domains. Reconstruction of these particle images with twofold symmetry imposed yielded a structure of dimeric ggCLC-7 at a resolution of 2.9 Å that enabled model building (*Figure 1A*, *Figure 1—figure supplement 2*, *Figure 1—figure supplement 3* and *Table 1*). The final refined model, which lacks the disordered N- and C-termini, fits well into the density with good geometry (*Figure 1B* and *Figure 1—figure supplement 2*, *Figure 1—figure supplement 3* and *Table 1*). Each protomer of dimeric ggCLC-7 contains a transmembrane domain composed of 18-transmembrane helices and a cytoplasmic domain composed of an N-terminal domain and two C-terminal cystathionine β-synthase (CBS) domains (*Figure 1C*). Both the transmembrane and cytosolic domains contribute to the large (~3700 Å$^2$) ggCLC-7 dimer interface (*Figure 1D*).

### Chloride and proton conduction pathways

The transmembrane domain of ggCLC-7 adopts the canonical CLC architecture with each protomer possessing discrete ion permeation pathways that extend from the cytosol to the lysosomal lumen (*Figure 2A*). Structural and functional analysis of CLC transporters and channels have defined the Cl--conduction pathway and its three conserved Cl--binding sites (*Accardi et al., 2004*; *Accardi and Miller, 2004*; *Accardi et al., 2005*; *Basilio et al., 2014*; *Chavan et al., 2020*; *Dutzler et al., 2002*; *Dutzler et al., 2003*; *Feng et al., 2010*; *Feng et al., 2012*; *Park et al., 2017*; *Park and MacKinnon, 2018*; *Picollo et al., 2012*; *Wang et al., 2019*; *Zdebik et al., 2008*), and this architecture is well-preserved in ggCLC-7. In the present conformation of ggCLC-7, constrictions too narrow to accommodate Cl- ions exist on both ends of the Cl--conduction pathway (*Figure 2B,C*). On the cytosolic side of the pathway between the central and internal Cl--binding sites, the side chains of Ser200, Tyr501 and Tyr598 form a constriction with a minimum radius of 0.6 Å (*Figure 2B*). The luminal side of the Cl--conduction pathway contains two additional constrictions (*Figure 2C*). The first constriction, which has a minimum radius of 0.7 Å is immediately adjacent to the external Cl--binding site and is formed by the side chains of Glu243, Ile511 and the backbone of Lys242 and Glu243. The second constriction (1.1 Å minimum radius) is at the luminal entrance and is formed by the side chains of Lys242, Glu467 and the backbone of Gly241. Together the three constrictions yield an occluded state for the transporter, sealing off the external and central Cl--binding sites from the cytosol and the lysosomal lumen.

The most dynamic residue in the ion transport pathways of CLC transporters is Glu$_{gate}$, whose conformation changes during the transport cycle. In previous structures of CLC transporters, the side chain carboxylic moiety of Glu$_{gate}$ has occupied four different positions: 'middle' where it occupies the central Cl--binding site, 'up' where it moves toward the extracellular vestibule, 'down' where it occupies the central Cl- binding site and most recently 'out' where it reaches away from the Cl--conduction pathway toward the H+-conduction pathway (*Chavan et al., 2020*; *Dutzler et al., 2002*; *Dutzler et al., 2003*; *Feng et al., 2010*; *Last et al., 2018*; *Figure 2—figure supplement 1*). In ggCLC-7, the Glu$_{gate}$ (Glu243) adopts the 'up' conformation, where it participates in establishing one of the luminal constrictions (*Figure 2D* and *Figure 2—figure supplement 1*). A non-protein density was resolved between Glu$_{gate}$ and Glu467 that we assigned as a water molecule. This water

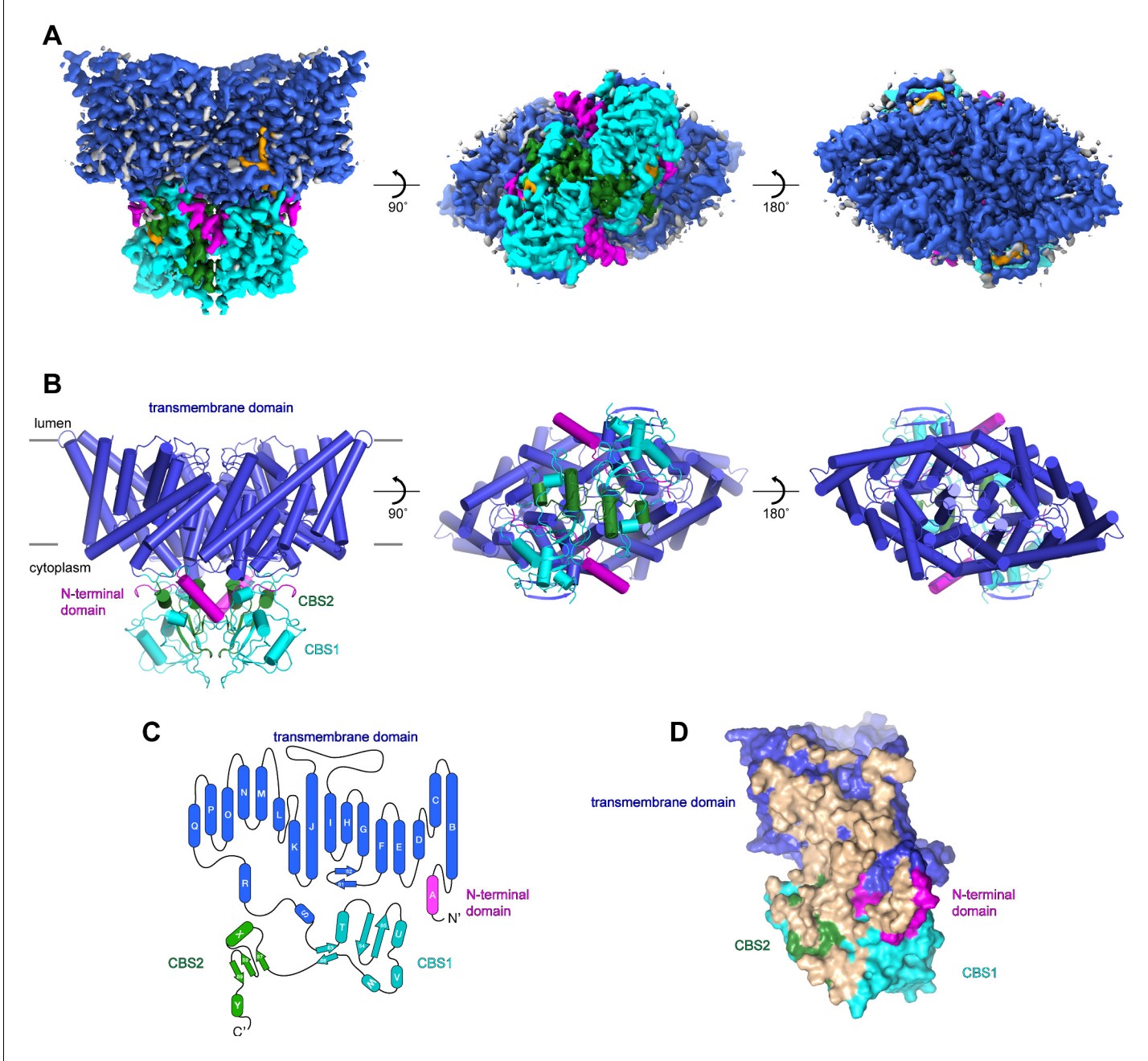

**Figure 1.** Structure of chicken CLC-7. (**A–B**) Cryo-EM density map (**A**) and structure (**B**) of ggCLC-7 viewed from within the membrane (left), the cytoplasm (middle) and the lysosomal lumen (right) colored by domain with N-terminal domain in magenta, transmembrane domain in blue, CBS1 in cyan and CBS2 in green. Modeled non-protein densities are colored orange and unmodeled non-protein densities are colored grey in A. (**C**) Domain topology of ggCLC-7 colored by domain as in A. (**D**) Dimer interface with interacting residues colored in wheat.

The online version of this article includes the following figure supplement(s) for figure 1:

**Figure supplement 1.** Cryo-EM analysis of human CLC-7.

**Figure supplement 2.** Cryo-EM analysis of chicken CLC-7.

**Figure supplement 3.** Representative cryo-EM density and model of chicken CLC-7.

may help to stabilize the conformations of $Glu_{gate}$ and Glu467, which may both be protonated at pH 8.0.

Within the $Cl^-$-conduction pathway, non-protein densities that we attributed to $Cl^-$ ions were resolved at the external, central and internal $Cl^-$-binding sites (*Figure 2D*). The external $Cl^-$ site is

**Table 1.** Cryo-EM data acquisition, reconstruction and model refinement statistics.

| | ggCLC-7 | hsCLC-7/OSTM1 | | | |
|---|---|---|---|---|---|
| | Consensus | Consensus | TMD Focus | LD Focus | CD Focus |
| **Cryo-EM acquisition and processing** | | | | | |
| EMDB accession # | 22386 | 22389 | | | |
| Magnification | 22,500x | 22,500x | 22,500x | 22,500x | 22,500x |
| Voltage (kV) | 300 | 300 | 300 | 300 | 300 |
| Total electron | 61 | 44 | 44 | 44 | 44 |
| Exposure ($e^-/Å^2$) | | | | | |
| Exposure time (s) | 8 | 4 | 4 | 4 | 4 |
| Defocus range (μM) | -1.0 to -2.5 | -1.0 to -2.5 | -1.0 to -2.5 | -1.0 to -2.5 | -1.0 to -2.5 |
| Pixel size (Å) | 1.0723 | 1.064 | 1.064 | 1.064 | 1.064 |
| Symmetry imposed | C2 | C2 | C1 | C1 | C1 |
| Initial particles | 4,020,225 | 15,288,379 | 15,288,379 | 15,288,379 | 15,288,379 |
| Final particles | 87,707 | 327,619 | 655,238 | 655,238 | 655,238 |
| Resolution (masked, Å) | 2.93 | 2.82 | 2.85 | 2.79 | 3.04 |
| Density modified CC (0.5, Å) | 2.92 | 2.81 | 2.80 | 2.70 | 3.30 |
| **Model refinement** | | | | | |
| PDB ID | 7JM6 | 7JM7 | | | |
| Model resolution (Å) | 2.98/2.40 | 2.92/2.56 | | | |
| FSC threshold | 0.50/0.143 | 0.50/0.143 | | | |
| Model refinement resolution | 300-2.9 | 300-2.8 | | | |
| **RMS deviations** | | | | | |
| Bond length (Å) | 0.005 | 0.005 | | | |
| Bond angle (°) | 0.806 | 0.769 | | | |
| **Ramachandran plot** | | | | | |
| Favored (%) | 97.73 | 97.87 | | | |
| Allowed (%) | 2.27 | 2.13 | | | |
| Disallowed (%) | 0 | 0 | | | |
| Rotamer Outliers (%) | 0.00 | 1.17 | | | |
| **Validation** | | | | | |
| MolProbity score | 1.37 | 1.37 | | | |
| Clashscore | 5.68 | 5.38 | | | |

formed by the backbone nitrogens of Glu243 and Gly244 on helix αF and Phe510 and Ile511 on helix αO. The intensity of the external Cl⁻-binding site (~14 σ) is the strongest of the three Cl⁻-binding sites and is nearly equivalent to that of backbone atoms of nearby residues, suggesting a high Cl⁻ occupancy. The density for the central Cl⁻ site has a slightly lower intensity (~12 σ) and is formed by the side chains of the highly conserved Ser200 from helix αD and Tyr598 on helix αS and the backbone nitrogens of Val509 and Phe510 of helix αN. The internal Cl⁻-binding site, which is located in a solvent-exposed vestibule on the cytoplasmic side of the transporter, has the lowest intermediate intensity (~8 σ) and is formed by the backbone nitrogens of Ser200 and Gly201 and the side chain of Gln204, all on helix αD. The relative intensities of the three Cl⁻-binding sites are consistent with structural and biochemical studies performed with *E. coli* CLC1 (ecCLC) that showed that the central and external binding sites have much higher affinity for Cl⁻ ions than the internal site (*Lobet and Dutzler, 2006*; *Picollo et al., 2009*).

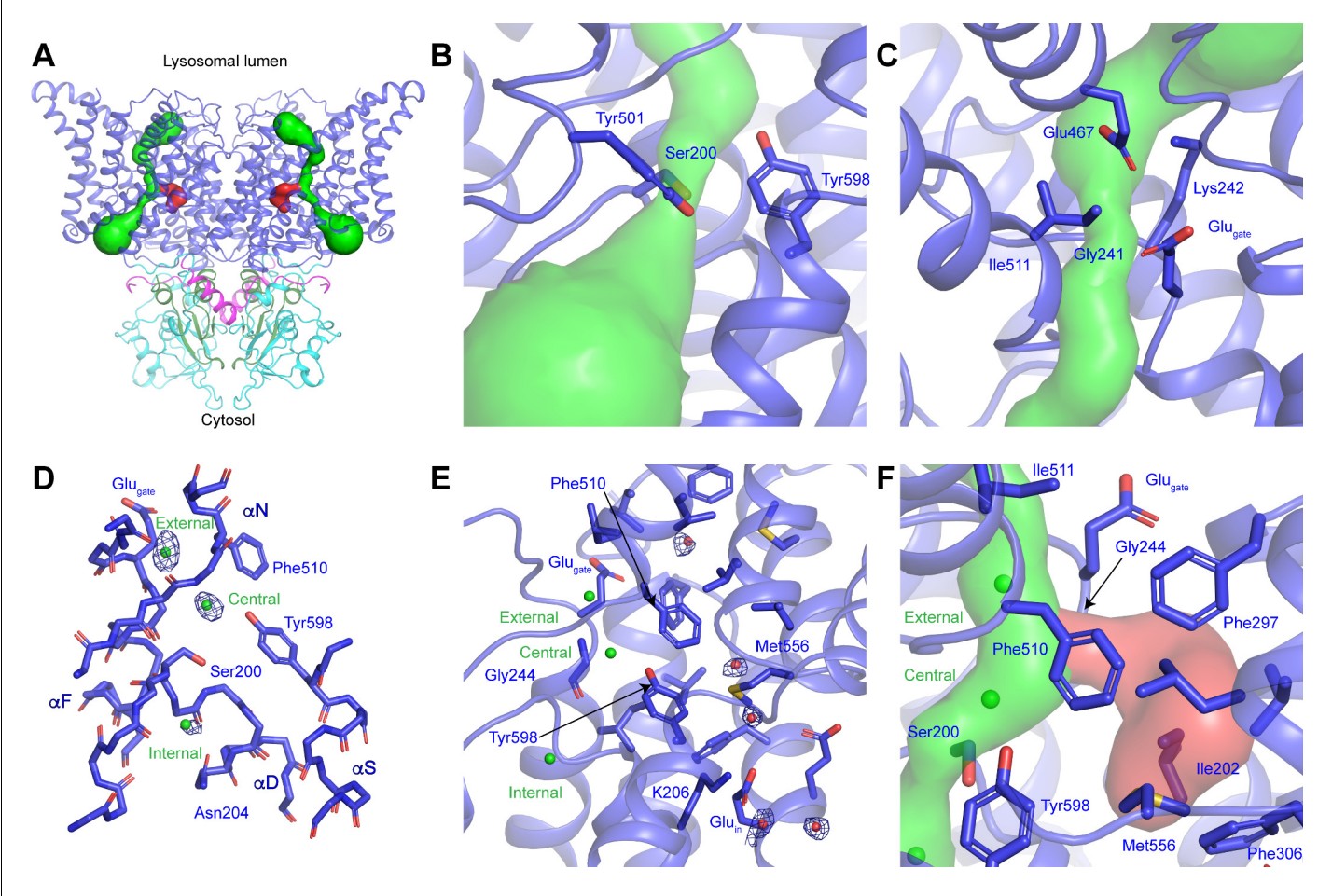

**Figure 2.** CLC-7 ion-conduction pathways. (**A**) Each protomer of ggCLC-7 contains a Cl⁻-conduction pathway, displayed as a green surface, and a putative H⁺-conduction pathway, displayed as a red surface. The N-terminal domain is colored in magenta, transmembrane domain in blue, CBS1 in cyan and CBS2 in green. (**B**) The cytosolic constriction of the Cl⁻-conduction pathway formed by Ser200, Tyr501 and Tyr598 narrows the pathway to a minimum radius of 0.6 Å. (**C**) Two constrictions exist near the luminal entrance to the pathway formed by $Glu_{gate}$ (Glu243), Lys242 and Ile511 and by Gly241, Lys242, Glu467. (**D**) Cl⁻-binding sites (shown as green spheres) in the ggCLC-7 ion conduction pathway. Experimental cryo-EM density is shown as blue mesh countered at 10 σ threshold. Conserved residues are shown in sticks. (**E**) Ordered water molecules (shown as red spheres) are resolved in the hydrophobic gap between $Glu_{gate}$ and $Glu_{in}$ and in the solvent-filled cavity in which $Glu_{in}$ resides. Experimental cryo-EM density is shown as blue mesh contoured at 7 σ threshold. (**F**) A potential H⁺-conduction pathway, shown as red surface, extends from near the central Cl⁻-binding site through into the hydrophobic gap. The access from the Cl⁻ pathway is lined by Gly244, Phe297 and Phe510. The pathway is separated from the cytosol by a constriction formed by Ile202, Phe306 and Met556.

The online version of this article includes the following figure supplement(s) for figure 2:

**Figure supplement 1.** Gating glutamate and the Cl⁻-conduction pathway.

Near the center of the transporter, the H⁺-conduction pathway of CLC transporters diverges from the Cl⁻-conduction pathway as they approach the cytosolic side of the transporter (*Accardi et al., 2005*; *Chavan et al., 2020*; *Han et al., 2014*; *Leisle et al., 2020*; *Park and MacKinnon, 2018*; *Wang and Voth, 2009*; *Zdebik et al., 2008*). This bifurcation occurs near the central Cl⁻-binding site and is proposed to extend through a hydrophobic gap to the conserved $Glu_{in}$ (*Chavan et al., 2020*). In the ggCLC-7 structure, $Glu_{in}$ (Glu310) on helix αG is located more than 15 Å away from $Glu_{gate}$, where it extends into a solvent filled cavity between the transmembrane and cytosolic domains that is continuous with the cytosol (*Figure 2E,F*). Within the loosely packed hydrophobic gap between $Glu_{gate}$ and $Glu_{in}$, several non-protein densities were resolved that we have tentatively modeled as water molecules (*Figure 2F*). Water molecules have previously been detected within the hydrophobic gap in structures and in molecular dynamics simulations of CLC transporters and have been

proposed to serve as a proton-conducting water-wire (*Chavan et al., 2020*; *Han et al., 2014*; *Leisle et al., 2020*; *Wang and Voth, 2009*). In ggCLC-7, the water molecules in the hydrophobic gap can access the $Cl^-$-conduction pathway through an opening with a minimum radius of ~1.4 Å between Gly244, Phe297 and Phe510 (*Figure 2F*). However, the pathway is not continuous with the cytosol as the hydrophobic gap is sealed near $Glu_{in}$ by a 1.0 Å constriction formed Ile202, Phe306 and Met558. In a recent structure of a mutant of ecCLC, the constrictions between $Glu_{gate}$ and $Glu_{in}$ were both expanded, creating a continuous pathway that would facilitate $H^+$ conduction (*Chavan et al., 2020*). It is possible that a similar conformational change may occur during the transport cycle of ggCLC-7 to open the constrictions and allow protons to pass through the hydrophobic gap.

## Organization of the CLC-7 cytoplasmic domain

The cytoplasmic domain of CLC-7 is composed of the N-terminal domain and the two C-terminal CBS domains (*Figure 3A*). The N-terminal domain, which has not been resolved in previous CLC structures, is comprised of a 14-amino acid extended segment and helix αA that are well-defined in the ggCLC-7 density map (*Figure 3B* and *Figure 1—figure supplement 3*). The extended segment is positioned at the center of the three-way interface between the transmembrane domain, CBS1 and CBS2. Because of its central position, the N-terminal domain is a major contributor to the tertiary and quaternary structure of ggCLC-7. Indeed, the N-terminal domain forms a larger interface with the transmembrane domain than either of the CBS domains.

Immediately adjacent to the N-terminal domain in a groove between the two CBS domains is a large density that cannot be attributed to the protein (*Figure 3C*). We modeled this non-protein density as a $Mg^{2+}$-bound ATP based on its shape and a comparison with the ATP-bound structure of the isolated CBS domains of CLC-5 (*Meyer et al., 2007*; *Figure 3—figure supplement 1*). Notably, no nucleotides were added during the 30-hour purification of ggCLC-7 so any ATP present must have been co-purified with the transporter. In the ggCLC-7 structure, the transporter forms multiple interactions with all three components of the ATP molecule (*Figure 3C*). The adenine base of the ATP is sandwiched between the side chains of His654 on CBS1, with which it forms π-stacking interactions, and Met778 on CBS2. The adenine base also forms polar interactions with the side-chain oxygen and backbone nitrogen of Thr632 and the backbone nitrogen of Gly656 that contribute to the specificity for adenine nucleotides (*Meyer et al., 2007*). The ribose sugar forms polar interactions with side chains of Ser628 and Asp783. The triphosphate group is coordinated by residues from both CBS domains as well as the N-terminal domain. The α-phosphate interacts with the side chain of His654 and the backbone oxygen of Asn653 of CBS1, the side chain of Lys782 of CBS2. Coordinating the α- and β-phosphate is a $Mg^{2+}$ that is partially coordinated by Glu91 of the N-terminal domain. Additionally, the β-phosphate also interacts with the side chain of Asn655 and the backbone oxygen of His654 of CBS1 and the side chain of His765 of CBS2. The γ-phosphate interacts with the side chain and backbone nitrogen of Ser92 of the N-terminal domain and the side chain of Arg764 of CBS2. Together, the numerous interactions between ATP and ggCLC-7 and the slow-off rate of ATP during the purification are consistent with ATP binding to ggCLC-7 with high affinity.

The ATP-binding site in ggCLC-7 shares many features with the ATP-binding site resolved in the structure of the isolated CBS domains of CLC-5, including the coordination of the adenine base and the ribose sugar (*Figure 3—figure supplement 1*; *Meyer et al., 2007*). The major differences between the ATP interactions of human CLC-5 and ggCLC-7 are the additional interactions with the triphosphate group of ATP in ggCLC-7. In particular, both ATP and $Mg^{2+}$ directly interact with the N-terminal domain of CLC-7 (*Figure 3C* and *Figure 3—figure supplement 1*). As the N-terminal domain was not present in the CLC-5 CBS domain crystallization construct (*Meyer et al., 2007*), such interactions were not previously identified. Binding studies performed with the CLC-5 CBS domains found that ATP binds with an affinity of ~100 µM, similar to affinities measured for ADP and AMP (*Meyer et al., 2007*). In contrast, the densities corresponding to the α-phosphate and γ-phosphate are nearly equivalent in the ggCLC-7 density map, indicating that ATP was predominant species co-purified with transporter (*Figure 3C*).

Sequence alignment of ggCLC-7 with the nine human CLC proteins reveals the N-terminal domain resolved in the ggCLC-7 structure (residues 87–114) is conserved in human CLC-7 as well as the closely-related human CLC-6 transporter (45% identity). The conservation is less clear in the more distantly related CLC-3, CLC-4 and CLC-5 transporters and no conservation is apparent in the

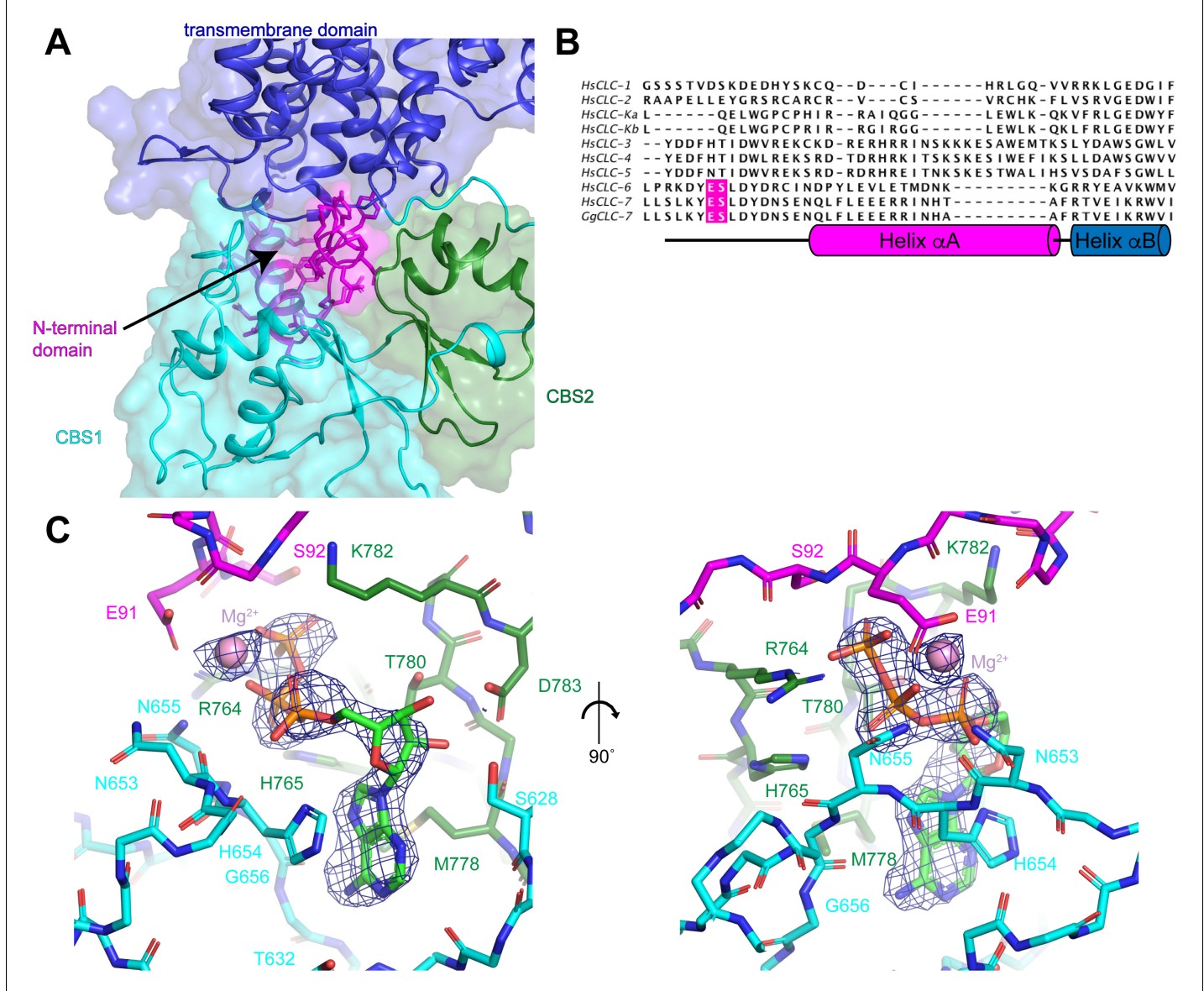

**Figure 3.** N-terminal domain and the ATP-binding site. (A) The N-terminal domain resides at the interface between the transmembrane domain, CBS1 and CBS2. N-terminal domain is colored in magenta, transmembrane domain in blue, CBS1 in cyan and CBS2 in green. (B) Sequence alignment of N-terminal domain of ggCLC-7 with human CLC-1, CLC-2, CLC-Ka, CLC-Kb, CLC-3, CLC-4, CLC-5, CLC-6, CLC-7. Positions of ATP coordinating Glu91 and Ser92 in ggCLC-7 are highlighted in magenta. (C) Two views of the ATP binding site in the cytoplasmic domain of ggCLC-7. Side chains that interact with ATP are shown as sticks. Experimental cryo-EM density is shown as blue mesh contoured at 12 σ threshold. Mg$^{2+}$ ion shown as a pink sphere.

The online version of this article includes the following figure supplement(s) for figure 3:

**Figure supplement 1.** Comparison of ATP-binding sites in ggCLC-7 and hsCLC-5.

CLC-1, CLC-2, CLC-Ka and CLC-Kb channels (*Figure 3B*). Furthermore, Glu91 and Ser92, the residues that interact with the triphosphate group of ATP in ggCLC-7, are only conserved in CLC-6 and CLC-7, indicating that ATP binding may vary among CLC proteins. We therefore speculate that ATP is the preferred ligand for CLC-7 and that further studies resolving the N-terminal domain of other CLC transporters will reveal to what extent CLC transporters bind specific adenine nucleotides.

## Phosphatidylinositol binding site

Non-protein densities that likely correspond to either ordered lipids or detergents were resolved around the periphery of the transmembrane domain of ggCLC-7 (*Figures 1A* and *4A,B*). Because it is difficult to distinguish lipids from detergents based on cryo-EM density maps alone, we were able to assign only one of the densities. The well-resolved head group allowed us to model the density as a phosphoinositol-3-phosphate (PI3P), which is a low-abundance constituent of lysosomal membranes (*Figure 4A*). Similar to ATP, PI3P was co-purified with the transporter from the HEK293S GnTI⁻ cell membranes. The PI3P molecule is located at the interface between the transmembrane domain and the cytosolic domain and interacts with residues from both domains. An amphipathic β-hairpin between helices αF and αG containing multiple positively charged residues surrounds the inositol phosphate head group and glycerol backbone, separating them from the rest of the membrane (*Figure 4A–C*). The head group also interacts with Lys213, Arg219 and Arg714, which

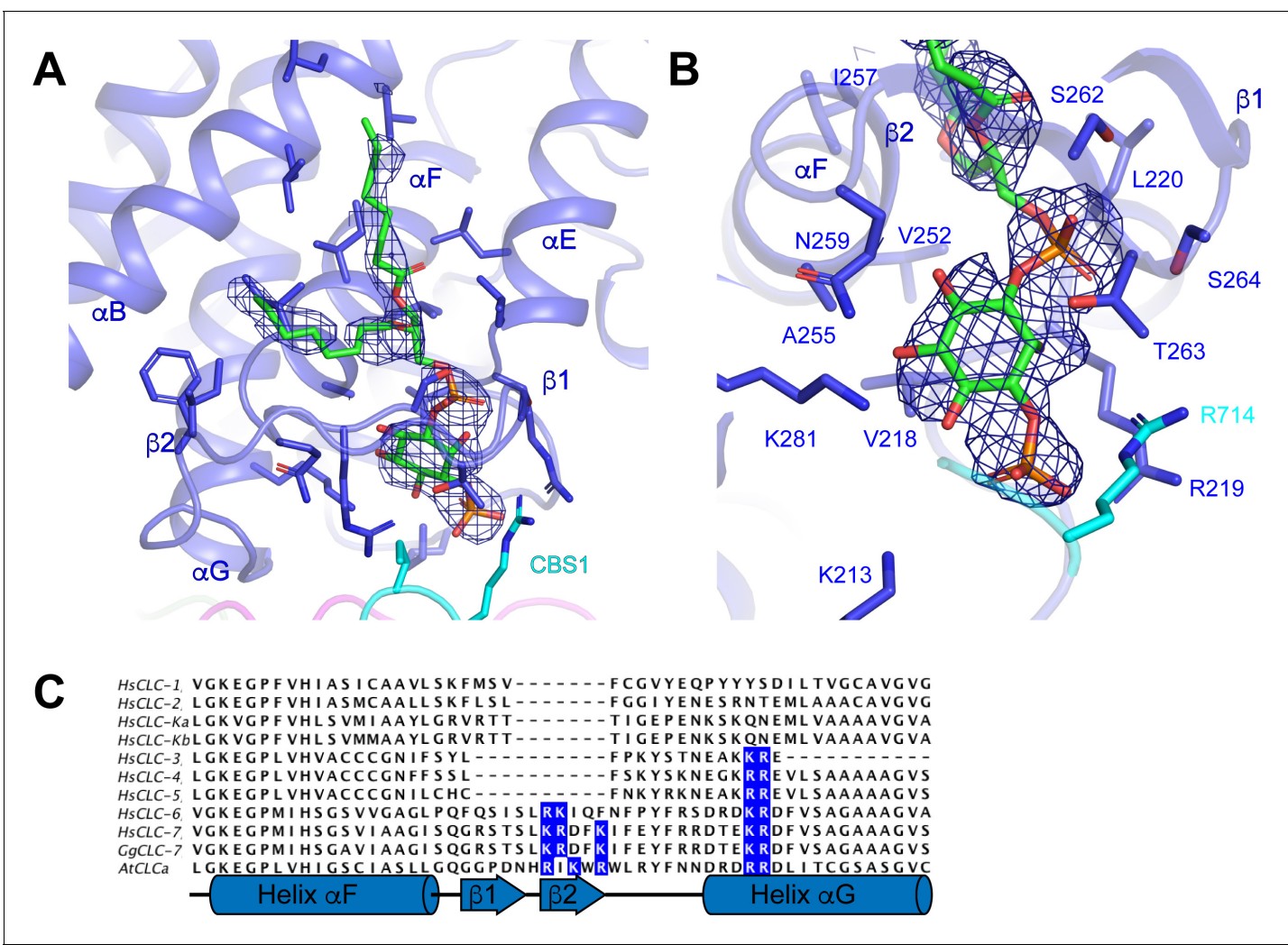

**Figure 4.** Phosphatidylinositol 3-phosphate binding site. (A) PI3P molecule shown as sticks. ggCLC-7 N-terminal domain is colored in magenta, transmembrane domain in blue, CBS1 in cyan and CBS2 in green with residues that interact with PI3P shown as sticks. Experimental cryo-EM density for PI3P is shown as blue mesh contoured at 10 σ threshold. (B) Coordination of the PI3P by ggCLC-7. Residues that interact with PI3P head group are shown as sticks. Experimental cryo-EM density for PI3P is shown as blue mesh contoured at 10 σ threshold. (C) Sequence alignment of helices αF and αG in ggCLC-7 with human CLC-1, CLC-2, CLC-Ka, CLC-Kb, CLC-3, CLC-4, CLC-5, CLC-6, CLC-7 and *A. thaliana* CLC-a. Positions of lipid coordinating Lys266, Arg267, Lys270, Lys281 and Arg282 in ggCLC-7 are highlighted in blue.

The online version of this article includes the following figure supplement(s) for figure 4:

**Figure supplement 1.** Modeling of phosphatidylinositol phosphate lipids into the PI3-binding site.

coordinate the phosphate group at the 3 position of inositol ring, and with Val218, Leu220, Val252, Ala255, Ser262, Thr263 and Lys281. Two 8-carbon acyl chains were modeled into PI3P density in a groove on the surface of the transporter formed by helices αB, αE and αF.

In ggCLC-7, PI3P is largely coordinated by helices αF and αG and the intervening β-hairpin (*Figure 4*). Among human CLCs, the elaborated loop between helices αF and αG present in ggCLC-7 is conserved in CLC-7 as well as CLC-6. The loop is also present in the vacuolar nitrate/H$^+$ antiporter CLC-a from *Arabidopsis thaliana* (atCLC-a) (*De Angeli et al., 2006*; *Figure 4C*). Based on the structure of ggCLC-7 and alignment of the sequences, three positively charged residues were identified within the β-hairpin of the transporters that may facilitate access of the negatively-charged lipid into the binding pocket (*Figure 4C*). In addition, several other residues that participate in the coordination of the PI3P head group, including Lys281 and Arg714 from ggCLC-7, are also conserved among hsCLC-6, hsCLC-7 and atCLC-a. Together, these data suggest that the PI-binding site may be a conserved feature among a subset of CLC transporters.

## Structure of the human CLC-7/OSTM1 complex

Unlike the other mammalian CLC transporters expressed in endosomes and lysosomes, CLC-7 is not active by itself. CLC-7 activity is dependent on the presence of its β-subunit, OSTM1 (*Lange et al., 2006*; *Leisle et al., 2011*). To better understand the role of OSTM1 in CLC-7-mediated Cl$^-$/H$^+$ exchange, we next co-expressed human CLC-7 and human OSTM1 in HEK293S GnTI$^-$ cells, purified the complex to homogeneity and analyzed its structure by cryo-EM. Similar to vitrified human CLC-7 by itself, human CLC-7/OSTM1 particles also adopted a preferred orientation in the ice. However, the effect was less severe for the CLC-7/OSTM1 particles and by collecting a large data set, we were able to resolve additional views. Two-dimensional and three-dimensional classification revealed the presence of intact CLC-7/OSTM1 complexes in the data set as well as a minor population of free CLC-7 dimers (*Figure 5—figure supplement 1*). Due to a low abundance and a preferred orientation of free CLC-7 particles, structural reconstitution of the CLC-7 homodimer was not possible. By employing a hierarchical classification approach, we were able to identify a population of intact CLC-7/OSTM1 complexes in which the cytoplasmic, transmembrane and luminal domains were all clearly resolved. Reconstruction of these particle images with two-fold symmetry imposed yielded a structure of CLC-7 in complex with OSTM1 at a resolution of 2.8 Å (*Figure 5A*, *Figure 5—figure supplement 1* and *Table 1*). 3D variability analysis of the selected particles revealed that the luminal domain of OSTM1 is flexibly attached to the transmembrane domain and adopts a range of different orientations. We observed up to a 6 Å displacement of the peripheral regions of the luminal domain of OSTM1 when the different states were aligned by their transmembrane domains (*Figure 5—figure supplement 2*). We therefore applied masks and performed local refinements, which yielded separate density maps at resolutions between 2.8 and 3.1 Å with improved interpretability for the transmembrane and cytosolic domains of CLC-7 and OSTM1 and the luminal domain of OSTM1 (*Figure 5—figure supplement 1* and *Table 1*). Despite the preferred orientation of the raw data set, the focus refined reconstructions determined with the selected particles display only minimal anisotropy and, following merging into a single composite map, were suitable for model building and coordinate refinement (*Figure 5B*, *Figure 5—figure supplement 1*, *Figure 5—figure supplement 3* and *Table 1*). The final refined structure contains two symmetrical copies of both CLC-7 and OSTM1 that fit well into the density with good geometry. When viewed from the side, the two copies of OSTM1 wrap around three sides of the CLC-7 dimer (*Figure 5*). The luminal domains of OSTM1 form a dimeric cap-like structure that covers the luminal surface of CLC-7 while the transmembrane helices pack against the periphery of the CLC-7 transmembrane domain. The C-terminal cytoplasmic domain of OSTM1 is disordered and no cytoplasmic interactions with CLC-7 were resolved.

The luminal domain of OSTM1 is composed of two three-helix bundles (*Figure 6A*). Connecting both within and between the helical bundles are five disulfide bonds that constrain the organization of the luminal domain. In the first bundle, a disulfide bond connects helix 1 to the short helix 3. In the second bundle, disulfide bonds connect helix 6 to helix 5 and to the linker between helix 7 and transmembrane helix 8. Between the two bundles, disulfide bonds connect the linker between helices 4 and 5 to helix 2 of the first bundle and to helix 7 of the second bundle (*Figure 6A,B*). The two helical bundles create a large dimer interface that buries ~4600 Å$^2$ of shared surface area (*Figure 6C*). The core of the dimer interface is formed by an antiparallel packing of helices 1 and 4 with helices 3 and 7 and several of the inter-helical linkers also making substantial contributions.

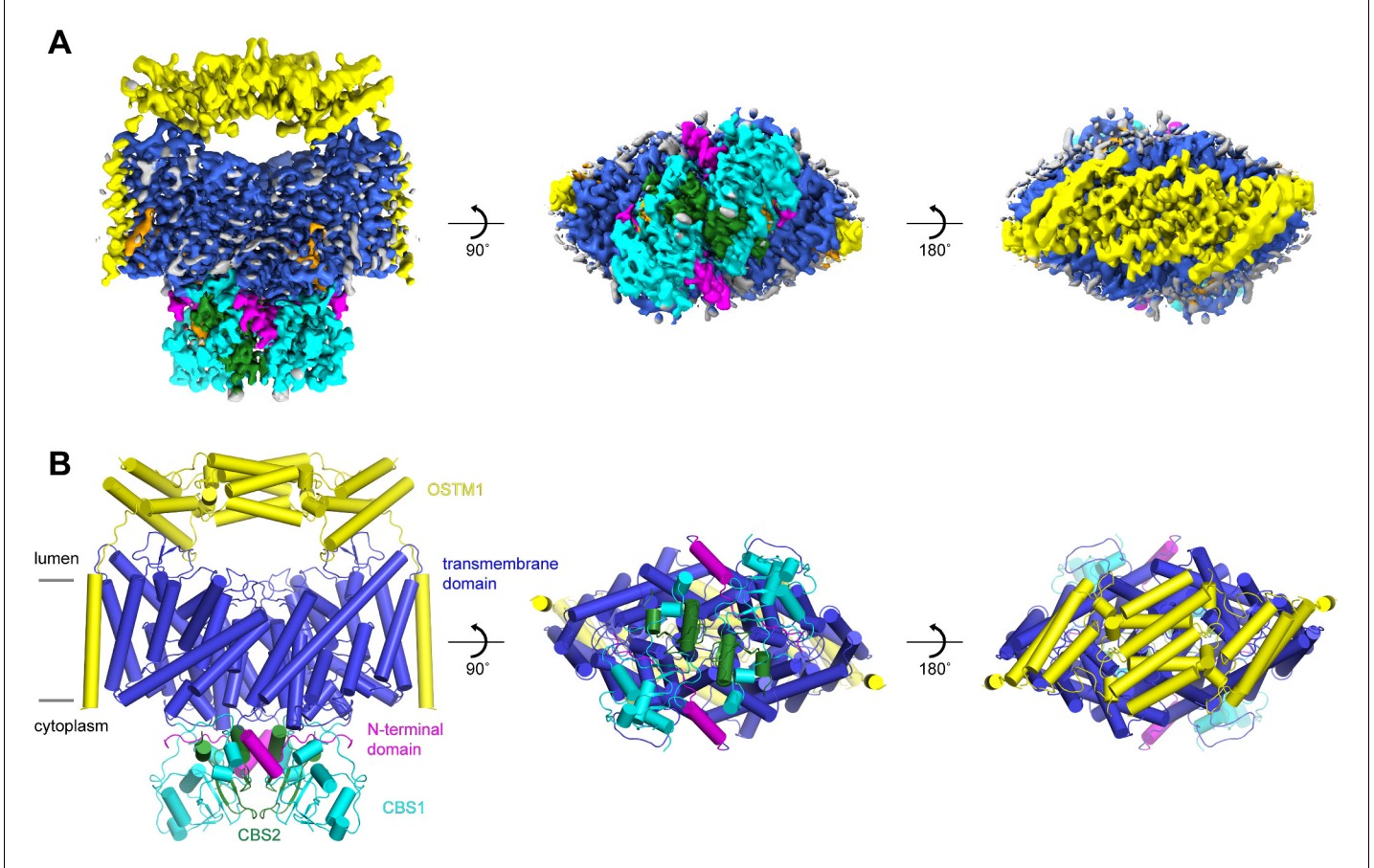

**Figure 5.** Structure of the human CLC-7/OSTM1 complex. (**A–B**) Cryo-EM density map (**A**) and structure (**B**) of CLC-7/OSTM1 complex viewed from within the membrane (left), the cytosol (middle) and the lysosomal lumen (right). CLC-7 is colored by domain with N-terminal domain in magenta, transmembrane domain in blue, CBS1 in cyan and CBS2 in green. OSTM1 is colored yellow. Modeled non-protein densities are colored orange and unmodeled non-protein densities are colored grey in **A**.

The online version of this article includes the following figure supplement(s) for figure 5:

**Figure supplement 1.** Cryo-EM analysis of human CLC-7/OSTM1.
**Figure supplement 2.** Three-dimensional variability analysis of human CLC-7/OSTM1.
**Figure supplement 3.** Representative cryo-EM density and model of human CLC-7/OSTM1.
**Figure supplement 4.** Comparison of ligand-binding sites in ggCLC-7 and human CLC-7/OSTM1.

While most of the interactions that stabilize the OSTM1 dimer interface are hydrophobic including the entirety of the helix 1-helix 1 and helix 3-helix 3 interactions, several polar interactions are also present including an ionic interaction between Arg107 and Asp150 (*Figure 6D*).

At the periphery of the luminal domain, non-protein densities were resolved extending from seven exposed asparagine residues (93, 128, 163, 184, 194, 263 and 274) on OSTM1 (*Figure 6A*). As previous computational analysis had identified these residues as consensus sites for N-linked glycosylation (*Lange et al., 2006*), we modeled these non-protein densities as carbohydrate moieties. The quality and interpretability of the carbohydrate densities varied between the seven sites, allowing us to model chains of different length. For example, density for a single N-linked N-acetyl-glucosamine group was resolved for Asn93 and Asn163, while a branched five-sugar carbohydrate moiety was resolved for Asn263 (*Figure 5—figure supplement 3*). While only minimal carbohydrate moieties can be added to N-linked glycosylation sites in the HEK293S GnTI⁻ cell line used for protein expression due to a mutation in N-acetyl-glucosaminyltransferase I, in non-glycoslylation-defective mammalian cells these glycosylation sites would be elaborately decorated and likely encase the entire surface of the luminal domain. Because CLC-7 lacks any N-linked glycosylation sites, the

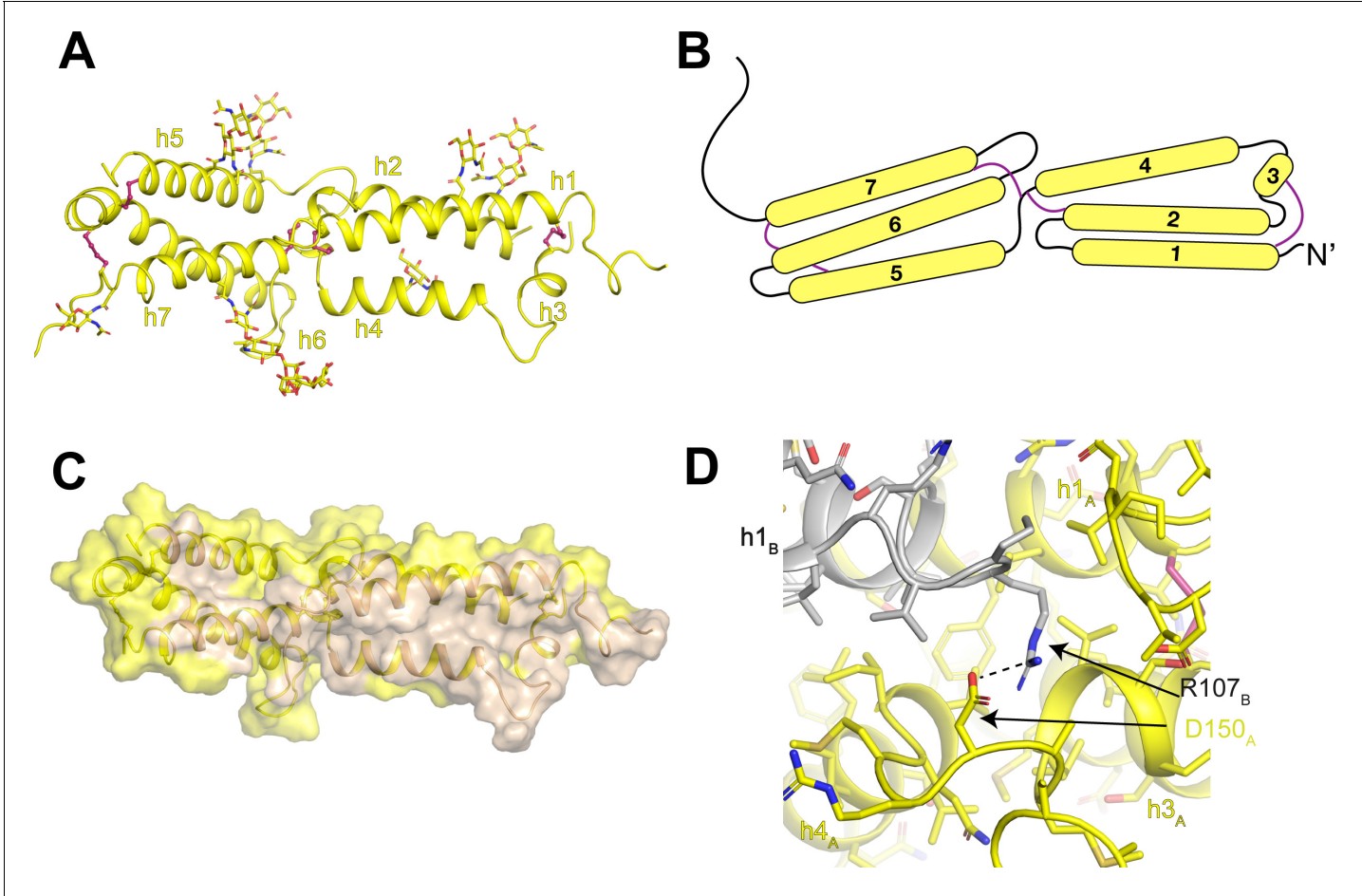

**Figure 6.** Architecture of the human OSTM1 luminal domain. (**A**) Monomeric structure of the hsOSTM1 luminal domain. Disulfide bonds are shown as pink sticks and glycosylated asparagine residues are shown as sticks. (**B**) Domain topology of hsOSTM1 with disulfides depicted as pink lines. (**C**) hsOSTM1 dimer interface. Residues that mediate inter-protomer interactions are colored in wheat. (**D**) Inter-protomer interaction between D150 of protomer A and R107 of protomer B. Protomer A is colored yellow and protomer B is colored grey.

glycosylation shell surrounding the rigid, disulfide-linked core of OSTM1 likely protects the luminal domain of CLC-7 from the harsh degradative environment of the lysosomal lumen.

## Effects of OSTM1 binding to CLC-7

We next compared the structure of ggCLC-7 with the structure of the human CLC-7/OSTM1 complex to determine how OSTM1 binding influences the conformation of CLC-7. Overall, the CLC-7 dimers show good alignment (RMSD 0.4 Å) in the presence and absence of OSTM1 (*Figure 7A*). The cytosolic domains and most of the transmembrane domains are essentially identical. Moreover, densities corresponding to ATP and PI3P molecules were resolved in their respective binding sites and the ligands interact with CLC-7 in a similar fashion regardless of the presence or absence of OSTM1 (*Figure 5A*, *Figure 5—figure supplement 3* and *Figure 5—figure supplement 4*). The only detectable rearrangements in CLC-7 occur near interfaces where CLC-7 directly contacts OSTM1 (*Figure 7A*). The largest CLC-7/OSTM1 interface is formed between helix 8 of OSTM1 and the transmembrane domain of CLC-7 (*Figure 7B*). Binding of OSTM1 is accompanied by a bend in helix αB of CLC-7 at Gly149 that results in 9 Å shift of the luminal end of helix αB (measured at Cα of Glu168) toward helix αK (*Figure 7—figure supplement 1*). Small (<2 Å) movements are resolved in the luminal ends of the nearby helices αC and αK toward OSTM1 helix 8. The rearrangements in helices αB, αC and αK of CLC-7 allow helix 8 of OSTM1, which is slightly kinked near Pro296, to pack against the surface of CLC-7 (*Figure 7B*). The interaction between transmembrane domains is largely mediated by the packing of hydrophobic residues, but polar interactions are formed between Tyr300 of

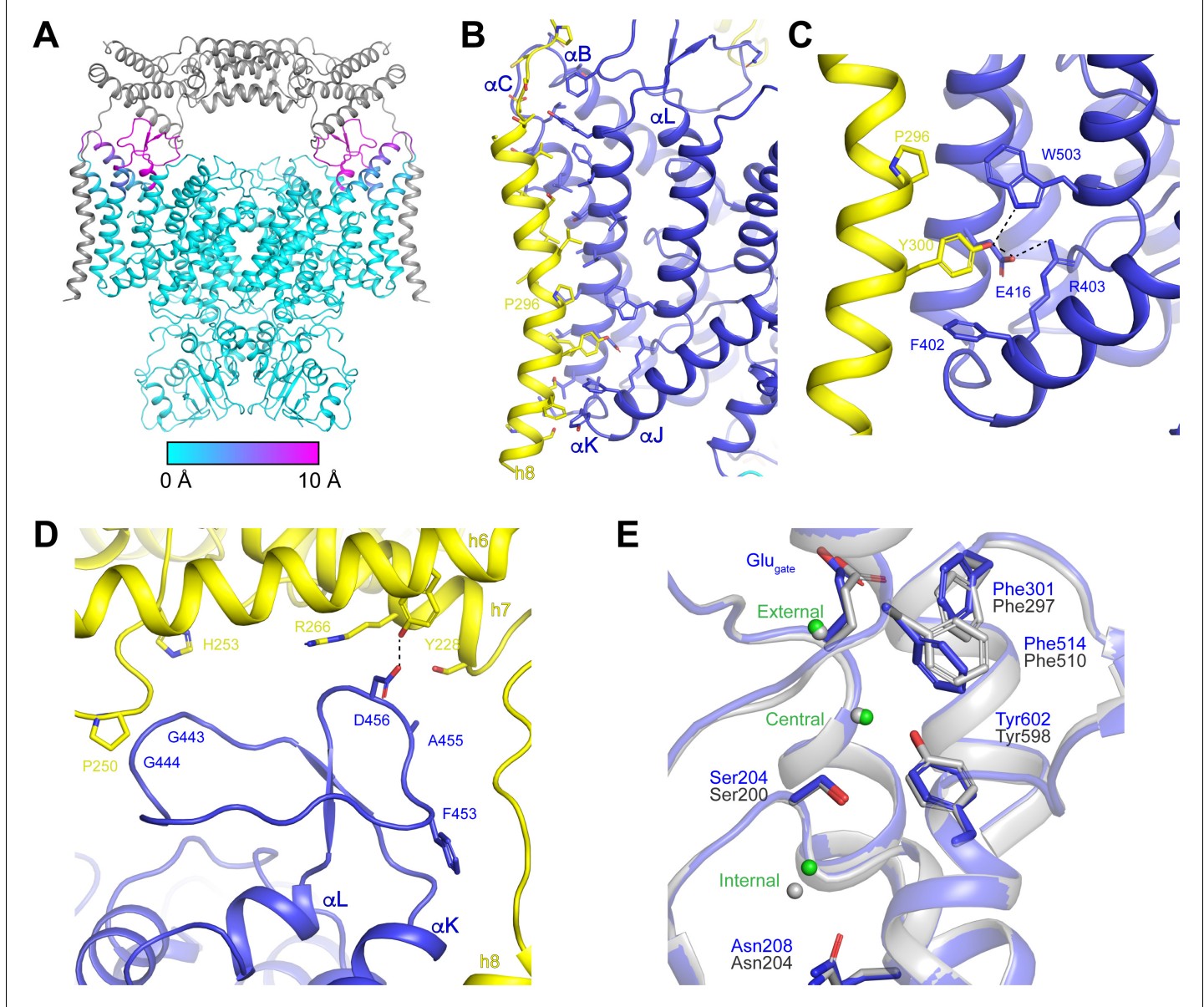

**Figure 7.** OSTM1-induced conformational changes. (**A**) Structure of hsCLC-7/OSTM1 with CLC-7 colored by Cα displacement compared to ggCLC-7 and OSTM1 colored in grey. (**B**) hsCLC-7/OSTM1 transmembrane domain interface. Residues that participate in the interaction are shown as sticks. CLC-7 is colored in blue and OSTM1 is colored yellow. (**C**) Polar interaction network in the transmembrane domain interface between CLC-7 and OSTM1. (**D**) Loop between αK and αL is stabilized by interactions with OSTM1 luminal domain. (**E**) Cl⁻-conduction pathways of human CLC-7/OSTM1 (colored by domain) and ggCLC-7 (grey). Interacting side chains are shown as sticks and Cl⁻ ions are shown as spheres. Blue residue numbers correspond to human CLC-7 and grey numbers correspond to ggCLC-7.

The online version of this article includes the following figure supplement(s) for figure 7:

**Figure supplement 1.** Structures of CLC-7 with and without OSTM1.

**Figure supplement 2.** Cl⁻-conduction pathway in human CLC-7/OSTM1.

**Figure supplement 3.** Heterogeneity in the conformation of Phe514/Phe510.

**Figure supplement 4.** Ion conduction pathways in human CLC-7/OSTM1.

OSTM1 and Glu416 and Trp503 of CLC-7 that may contribute to the specificity of the CLC-7/ OSTM1 interaction (*Figure 7C*). OSTM1 also contacts and stabilizes the linker between helices αK and αL, which was too poorly ordered to be modeled in absence of OSTM1 in the ggCLC-7 density map (*Figure 7A,D*). In the CLC-7/OSTM1 complex, the 25-residue linker between helices αK and αL

forms two interactions with the luminal domain of OSTM1. Asp456 of CLC-7 forms a polar interaction with Tyr228 of helix 6. Gly443 and Gly444 of CLC-7 form a small interface with a portion of the loop between helices 6 and 7 that includes Pro250, Gly251 and His253. In addition to the direct protein-protein interactions, a non-protein density that may correspond to a cholesterol based on its size and shape was resolved at the interface between the transmembrane domains of CLC-7 and OSTM1 (*Figure 5—figure supplement 3*). Notably, no density was present at this site in the ggCLC-7 map, suggesting that CLC-7 and OSTM1 together may form an additional lipid-binding site.

To examine the influence of OSTM1 on $Cl^-$ and $H^+$ transport, we compared the ion conduction pathways in the CLC-7/OSTM1 and ggCLC-7 structures (*Figure 2*, *Figure 7E*, *Figure 7—figure supplement 2* and *Figure 7—figure supplement 4*). In both structures, the $Cl^-$-conduction pathways adopt similar occluded states with narrow constrictions present at either end and ions occupying the three binding sites. Superpositioning reveals that most of the pore-lining residues are positioned similarly in the presence and absence of OSTM1. The only residues that adopt differing conformations are Phe301 (Phe297 in ggCLC-7) and Phe514 (Phe510 in ggCLC-7), both of which are highly conserved among CLC family members and whose mutation in human CLC-7 leads to defects in $Cl^-$/ $H^+$ exchange coupling, voltage-dependence and activation (*Leisle et al., 2020*). Inspection of the CLC-7/OSTM1 density map reveals that the side-chain density for Phe514 is distorted in a manner consistent with the side chain adopting both the modeled conformation (conformation 1) as well as a conformation similar to that resolved for Phe510 in ggCLC-7 (conformation 2) (*Figure 7—figure supplement 3*). Inspection of the ggCLC-7 density map revealed no density consistent with the alternative rotamer, indicating that conformation two is the predominant state for Phe510. While the existence of the two conformations for Phe514 in CLC-7/OSTM1 is clearer in a map sharpened to emphasize the high-resolution features, due to the limited resolution and the anisotropy present in the data, it is difficult to estimate occupancy of the different rotamers. Thus, while we modeled Phe514 as the predominant conformation 1, the data also supports the existence of conformation 2. Because Phe514 is located at the interface between the hydrophobic gap and the central $Cl^-$ binding site, changes in its conformation can modify the $H^+$-conduction pathway. In CLC-7/OSTM1, the side chain of Phe514 (conformation 1) narrows the constriction of this pathway to a minimum radius of 0.8 Å, which is too narrow to allow water molecules to enter the hydrophobic gap (*Figure 7—figure supplement 4*). In contrast, the alternative conformation adopted by Phe510 in the ggCLC-7 structure widens the pathway sufficiently to allow water molecules to cross (1.4 Å minimum radius) (*Figure 2F*).

We next compared the relative intensities of the ion binding sites to assess the effect on OSTM1 binding on $Cl^-$ binding in the permeation pathway. In ggCLC-7, the external and central sites exhibit strong densities that are only slightly weaker than nearby protein atoms, indicating a high occupancy for $Cl^-$ ions at these sites, while the density at the internal $Cl^-$binding site is significantly weaker (*Figure 2D*). The relative order of intensities differs in the CLC-7/OSTM1 structure (*Figure 7—figure supplement 2*). In CLC-7/OSTM1, the density at the external site is the strongest and has a similar intensity as nearby protein atoms (~14 σ). However, unlike in ggCLC-7, the central site in CLC-7/ OSTM1 is the weakest and is only slightly above the background (~4 σ). While we must be cautious in interpreting the densities occupying the $Cl^-$-binding sites of CLC-7/OSTM1 because of its anisotropic nature, the differences in relative intensities of the $Cl^-$-binding site peaks between ggCLC-7 and CLC-7/OSTM1 suggest that there may be a change in $Cl^-$ occupancy of the central site when CLC-7 is bound to OSTM1. A change in occupancy of the central $Cl^-$ site may be associated with the different conformation of Phe510/Phe514, which is located ~4 Å from the central $Cl^-$ site in both structures. Such as association would be consistent with molecular dynamics simulations performed using ecCLC that identified a coupling between $Cl^-$ occupancy at the central site and the conformation of Phe297, which is equivalent to Phe514 in human CLC-7 (*Leisle et al., 2020*).

Together, these data indicate that OSTM1 binding does not greatly perturb the conformation of the ion conduction pathways in CLC-7 and that its influence on CLC-7 transport activity does not occur through large-scale rearrangements. Rather, these data suggest that OSTM1 binding can potentially induce subtle conformational changes in key residues and provide critical structural support for CLC-7. Moreover, by virtue of its heavy glycosylation, OSTM1 can protect the un-glycosylated CLC-7 from degradation in the acidic lysosomal lumen.

## Discussion

In this study, we present structures of the lysosomal $Cl^-/H^+$ exchanger CLC-7 alone and in complex with its obligatory β-subunit OSTM1. The structure of the CLC-7/OSTM1 complex reveals that OSTM1 forms a heavily-glycosylated cap that covers the luminal surface of CLC-7 (*Figures 5* and *6*). OSTM1 associates with CLC-7 largely through interactions mediated by the transmembrane domains, consistent with analyses that demonstrated that deletion of the transmembrane domain of OSTM1 phenocopies the *Ostm1* null in mice (*Pata and Vacher, 2018*). When complexed with CLC-7, OSTM1 does not adopt the structure of a RING finger domain as had previously been suggested (*Fischer et al., 2003*). Instead, the luminal domain of OSTM1 forms a tightly packed core composed of helical bundles linked together by numerous disulfide bonds (*Figure 6*). This stable core, together with the glycosylated periphery make the luminal domain of OSTM1 well-suited to survive the harsh degradative environment of the lysosomal lumen. In contrast, CLC-7 alone among the human CLC transporters lacks any N-glycosylation sites of its own and is consequently unstable in the lysosome when expressed in the absence of OSTM1 (*Lange et al., 2006*). The structure of CLC-7/OSTM1 is thus consistent with OSTM1 serving a protective role to shield CLC-7 from proteolysis and degradation.

Comparison of the CLC-7 structures in the presence and absence of OSTM1 reveals that OSTM1 binding induces subtle changes to the conformations of the ion permeation pathways. Among the changes are the conformations of residues essential for proper transport activity and the occupancies of the $Cl^-$-binding sites. Notably, these conformational changes appear to occur only in a subset of the CLC-7/OSTM1 complexes. It is therefore possible that OSTM1 binding alters the equilibrium between different CLC-7 conformations. However, the current data do not enable accurate modeling of alternative rotamers, and thus it is not possible to compare the fraction of CLC-7 transporters adopting each possible state in the presence and absence of OSTM1. Moreover, we do not know precisely to which functional state these conformations correspond. In the CLC-7/OSTM1 and ggCLC-7 structures, the $Cl^-$-conduction pathways resemble those of the ecCLC E148Q mutant where the three $Cl^-$-binding sites are occupied and the $Glu_{gate}$ adopts the 'up' conformation where it can potentially exchange protons with the lumen (*Dutzler et al., 2003*). Because the 'up' conformation is a coordinate along the proposed transport cycle (*Feng et al., 2012*), and because CLC-7 functionality has been previously detected in the absence of OSTM1 using solid-supported membranes and in plant vacuoles (*Costa et al., 2012*; *Schulz et al., 2010*), it is possible that CLC-7 structures both in the presence and absence of OSTM1 represent states that are competent for $Cl^-/H^+$ transport activity. Notably, even in conditions where CLC-7 activity could be detected without OSTM1, current levels were significantly increased by its co-expression, consistent with OSTM1 potentiating transport activity (*Schulz et al., 2010*). Future investigations will thus be necessary to precisely determine how OSTM1 stimulates CLC-7 activity - whether by inducing conformational changes in the ion conduction pathways or merely stabilizing lysosomal expression of CLC-7.

Initial characterizations of CLC-7/OSTM1 demonstrated that its activity is dependent on membrane potential and luminal pH (*Leisle et al., 2011*; *Ludwig et al., 2013*). Our structures reveal the presence of ATP and phosphatidylinositol-binding sites, suggesting that additional signals may also regulate CLC-7 activity. Indeed, ATP has been demonstrated to influence the activity of multiple CLCs, but the precise effect of ATP on transporter activity has been controversial with evidence supporting both stimulatory and inhibitory roles. For example, addition of ATP increased transporter activity of CLC-4 by two-fold but inhibited activity of CLC-2 channels (*Stölting et al., 2013*; *Vanoye and George, 2002*). Moreover, the particular adenine nucleotide species that can influence CLC activity has been unclear. Binding studies conducted with the CBS domains of CLC-5 detected affinities of ~100 µM for ATP, ADP and AMP (*Meyer et al., 2007*). Recent studies revealed that CLC-3, CLC-4 and CLC-5 are able to distinguish between different nucleotide moieties, and showed that $Mg^{2+}$ ions modify the effect of ADP binding (*Grieschat et al., 2020*). In the structures of ggCLC-7 and CLC-7/OSTM1, we observe direct coordination of all three phosphates not only through interactions with the CBS domains but also with the N-terminal domain (*Figure 3* and *Figure 3—figure supplement 1*). Based on our data, we suggest that the actual binding affinity of CLC-7 to ATP is much higher than that detected for the CBS domains alone. As ATP levels are in excess of 1 mM under physiological conditions, it is likely that ATP is constitutively bound to CLC-7 and may therefore serve a structural role rather than a regulatory role. While ATP is a regulatory factor

for numerous proteins, a structural role for nucleotides has been previously described for AMP-activated protein kinase (AMPK), inositol 1,4,5-trisphosphate (IP3) receptors and some prokaryotic regulator of potassium conductance (RCK)-gated channels (*Bezprozvanny and Ehrlich, 1993*; *Cao et al., 2013*; *Hardie and Hawley, 2001*; *Kong et al., 2012*; *Kröning et al., 2007*; *Teixeira-Duarte et al., 2019*). Among the CLC family, the residues in the N-terminal domain of CLC-7 that interact with ATP and Mg$^{2+}$ are not broadly conserved, suggesting that the structural divergence of the ATP-binding site may contribute to varied effects that been reported among CLC family members.

Dysregulated ATP binding to CLC-7 may play a role in human disease. While CLC-7 remains fully functional in the absence of ATP (*Leisle et al., 2011*), mapping disease mutations onto CLC-7/OSTM1 reveals a hotspot of mutations on CBS2 near the ATP-binding site (*Figure 8*). Several distinct mutations of Arg767, which directly participates in the ATP γ-phosphate coordination, as well as mutations of neighboring Gly765 and Leu766 have been identified as leading to osteopetrosis (*Cleiren et al., 2001*; *Leisle et al., 2011*; *Sartelet et al., 2014*). Previous work characterizing the function of the Arg767 mutants revealed distinct phenotypes, with the R767P and R767W mutants

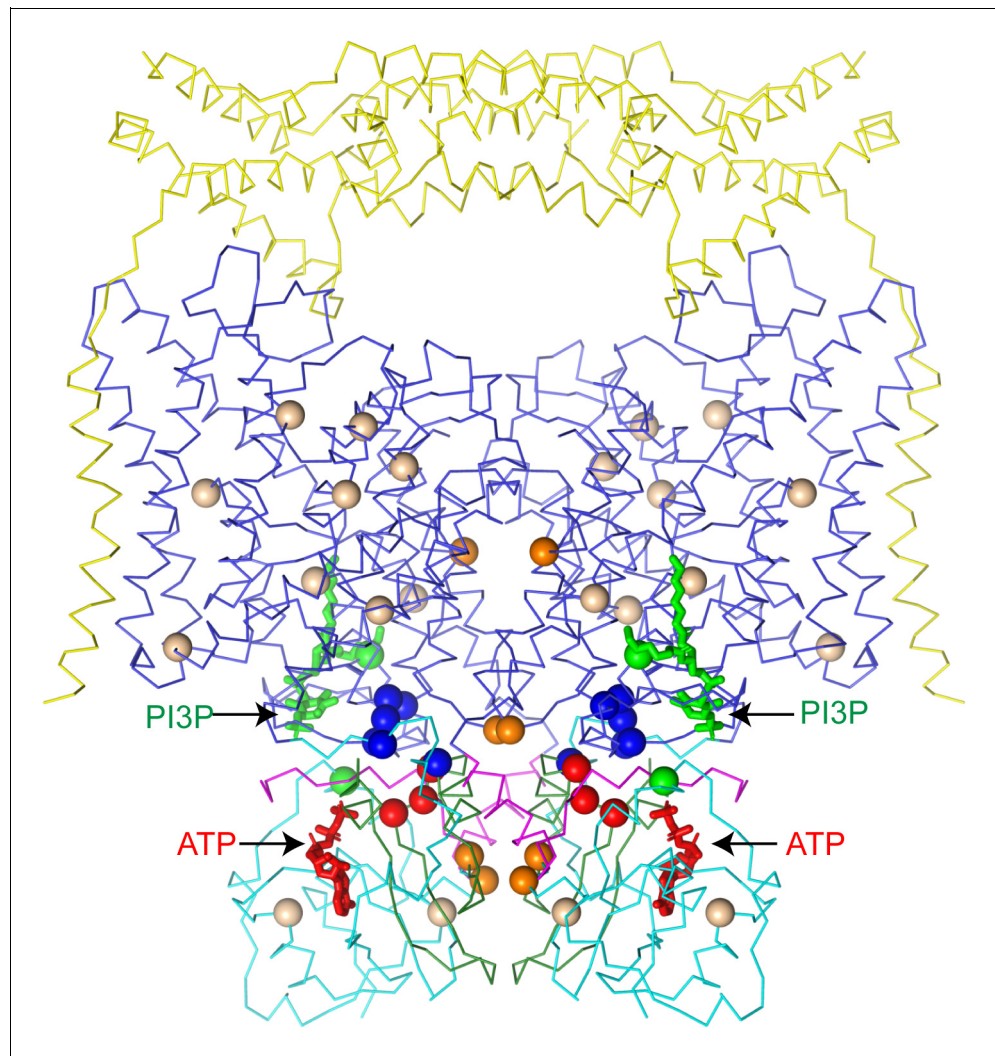

**Figure 8.** CLC-7 mutations associated with osteopetrosis and lipid storage disease. Structure of hsCLC-7/OSTM1 colored by domain with PI3P and ATP shown as green and red sticks, respectively. Spheres represent residues with disease-causing mutations. Wheat spheres represent residues with buried side chains, orange spheres represent residues at inter-domain interfaces, green spheres represent residues near the PI3P binding site, blue spheres represent residues at transmembrane-CBS-domain interfaces and red spheres represent residues near the ATP-binding site.

displaying no activity while the R767Q mutants displayed increased activation kinetics (*Leisle et al., 2011*). Together, these results indicate that while ATP may be constitutively bound and serve a structural role, disruption of the binding site has functional consequences for CLC-7.

The second ligand co-purified with ggCLC-7 and CLC-7/OSTM1 is PI3P, a phosphatidylinositol (PI) lipid species enriched in endolysosomal membranes that constitutes between 0.1% and 0.5% of the total PI content in cells (*Falasca and Maffucci, 2006*). While PI3P has not been previously characterized as a regulator of ion transport protein activity, the related phosphatidylinositol 3,5-bisphosphate (PI(3,5)$P_2$) is potent modulator of ion transport proteins whose abundance is tightly regulated (*Fine et al., 2018*; *Hasegawa et al., 2017*; *She et al., 2018*; *She et al., 2019*). Under basal conditions, PI(3,5)$P_2$ concentrations are very low (<0.1% of cellular PI content), but in yeast can rise more than 20-fold upon hyperosmotic shock (*Duex et al., 2006*). In endosomes and lysosomes, PI(3,5)$P_2$ binding activates the $Ca^{2+}$ channel TRPML1 (*Dong et al., 2010*) and the $Na^+$ channel TPC1 (*Wang et al., 2012*). In plant vacuoles, which share many features with lysosomes, PI(3,5)$P_2$ potently regulates atCLC-a, inhibiting its activity with an $IC_{50}$ of ~10 nM (*Carpaneto et al., 2017*). Because the PI3P-binding site appears to be conserved between atCLC-a and CLC-7, we modeled in a PI(3,5)$P_2$ lipid into the binding site in the CLC-7/OSTM1 structure to gain insights into its effect (*Figure 4—figure supplement 1*). In the model, a phosphate at 5-position of inositol ring could not be accommodated due to steric clashes with Lys281 on helix αG. We therefore speculate that binding of the regulatory PI(3,5)$P_2$ may induce conformational changes to CLC-7 that may alter its activity, analogous to the inhibitory effect of PI(3,5)$P_2$ on atCLC-a. Consistent with the PI lipids influencing transporter activity, a mutation of Tyr715, which is located near the PI3P-binding site, to cysteine was recently identified in the gene encoding human CLC-7 that causes a novel lipid storage disease without osteopetrosis (*Undiagnosed Diseases Network et al., 2019*). Functional analysis of this mutant revealed that it displays increased current levels when expressed in oocytes compared to wild-type CLC-7 and leads to a hyper-acidification phenotype in lysosomes (*Figure 8*; *Undiagnosed Diseases Network et al., 2019*).

For decades CLC-7 was perhaps the most enigmatic CLC family member (*Brandt and Jentsch, 1995*). Functional and structural characterization was limited by its lysosomal expression and its absolute requirement for a β-subunit, OSTM1. OSTM1 has been shown to have dual functions, both stabilizing CLC-7 in the lysosome and serving as an essential activator of the transporter (*Lange et al., 2006*; *Leisle et al., 2011*; *Stauber and Jentsch, 2010*). Our studies reveal how OSTM1 interacts with CLC-7 protecting the transporter from the acidic environment of the lysosomal lumen and lay the groundwork for future studies to elucidate how it, as well as ATP and the lipids identified in the structures, regulate CLC-7 transport activity and contribute to pH homeostasis in the lysosome and osteoclasts.

## Materials and methods

### Key resources table

| Reagent type (species) or resource | Designation | Source or reference | Identifiers | Additional information |
|---|---|---|---|---|
| Gene (*Gallus gallus*) | *ggCLC-7* | Synbio technologies | | |
| Gene (*Homo sapiens*) | *hsCLCN7* | Synbio technologies | | |
| Gene (*Homo sapiens*) | *hsOSTM1* | Synbio technologies | | |
| Cell line (*Homo sapiens*) | HEK-293T | ATCC | CRL-3216 RRID:CVCL_0063 | |
| Cell line (*Homo sapiens*) | HEK-293S GnTi- | ATCC | CRL-3022 | |

*Continued on next page*

*Continued*

| Reagent type (species) or resource | Designation | Source or reference | Identifiers | Additional information |
|---|---|---|---|---|
| Chemical compound, drug | Polyethylenimine, Linear, MW 25000, Transfection Grade (PEI 25K) | Polysciences, Inc | 23966–1 | |
| Chemical compound, drug | Sodium Butyrate | Sigma | 8451440100 | |
| Chemical compound, drug | Valproic acid | Sigma | P4543 | |
| Chemical compound, drug | Lauryl maltose neopentyl glycol | Anatrace | NG310 | |
| Software, algorithm | MotionCor2 | *Zheng et al., 2017* | RRID:SCR_016499 | |
| Software, algorithm | CtfFind 4.1.10 | *Rohou and Grigorieff, 2015* | RRID:SCR_016731 | |
| Software, algorithm | RELION 3.1 | *Scheres, 2016* | http://www2.mrc-lmb.cam.ac.uk/relion RRID:SCR_016274 | |
| Software, algorithm | SerialEM | *Mastronarde, 2005* | RRID:SCR_017293 | |
| Software, algorithm | cryoSPARC v2 | Structura Biotechnology | https://cryosparc.com/ RRID:SCR_016501 | |
| Software, algorithm | PHENIX | *Liebschner et al., 2019* | https://www.phenix-online.org/ RRID:SCR_014224 | |
| Software, algorithm | COOT | *Emsley et al., 2010* | https://www2.mrc-lmb.cam.ac.uk/personal/pemsley/coot/ RRID:SCR_014222 | |
| Software, algorithm | PyMOL | *Schrödinger LLC, 2020* | https://pymol.org/2/ RRID:SCR_000305 | |
| Software, algorithm | MOLE | *Pravda et al., 2018* | https://mole.upol.cz RRID:SCR_018314 | |
| Software, algorithm | UCSF Chimera | *Pettersen et al., 2004* | https://www.cgl.ucsf.edu/chimera RRID:SCR_004097 | |
| Software, algorithm | Blocres/Bsoft | Heymann | | |
| Software algorithm | Jalview | *Waterhouse et al., 2009* | https://www.jalview.org RRID:SCR_006459 | |
| Others | QUANTIFOIL R1.2/1.3 holey carbon grids | Quantifoil | | |
| Others | FEI Vitrobot Mark IV | FEI Thermo Fisher | | |

## Protein expression and purification

The gene encoding *CLCN7* from *Gallus gallus* was synthesized (SynBio) and subcloned into a Bac-Mam expression vector with a C-terminal mEGFP-tag fused via a short linker containing a PreScission protease site (*Goehring et al., 2014*). The plasmid was mixed with PEI 25K (Polysciences, Inc) at a 1 : 3 ratio for 30 min and then used to transfect HEK293S GnTi⁻ cells (ATCC: CRL-3022). For a 1 L cell culture 1 mg plasmid and 3 mg PEI 25K were used. After 24 hr incubation at 37 ˚C, sodium butyrate (Sigma) was added to a final concentration of 10 mM, and cells were allowed to grow at 37 ˚C for an additional 72 hr before harvesting. Cell pellets were washed in phosphate-buffered saline solution and flash frozen in liquid nitrogen. Membrane proteins were solubilized in 2% lauryl maltose neopentyl glycol (LMNG, Anatrace), 0.2% cholesteryl hemisuccinate tris salt (CHS, Anatrace), 20 mM HEPES pH 7.5, 150 mM KCl supplemented with protease-inhibitor cocktail (1 mM PMSF, 2.5 µg/mL aprotinin, 2.5 µg/mL leupeptin, 1 µg/mL pepstatin A) and spatula of DNaseI for 1 hr. Solubilized

proteins were separated by centrifugation 75,000 $g$ for 40 mins, followed by binding to 2.5 ml anti-GFP nanobody resin for 1 hr, which was equilibrated with washing buffer containing 0.1% LMNG, 50 mM Tris-HCl pH 8, 150 mM KCl, 2 mM DTT (BufferA). Anti-GFP nanobody affinity chromatography was performed by 20 column volumes of washing with BufferA, followed by overnight PreScission digestion, and elution with wash buffer. Protein sample was concentrated to a volume of 500 µl using CORNING SPIN-X concentrators (100 kDa cutoff), followed by centrifugation 10,000 g for 10 mins. Concentrated proteins were further purified by size exclusion chromatography on a Superose 6 Increase 10/300 GL (GE healthcare) in BufferA. Peak fractions were pooled and concentrated to ~2 mg/mL using CORNING SPIN-X concentrators (100 kDa cutoff).

Genes encoding human *CLCN7* and *OSTM1* were synthesized (SynBio) and subcloned into Bac-Mam expression vectors with C-terminal mCerulean- and mVenus- tags, respectively, fused via a short linker containing a PreScission protease site (*Goehring et al., 2014*). Transient transfection was carried out as described above for chicken *CLCN7*, with a single modification; for gene expression, valproic acid (VPA, Sigma) was added to induce expression at a final concentration of 2.2 mM. Equal amounts of plasmids encoding *CLCN7* and *OSTM1* were added to the reaction mix. Cell pellets were washed in phosphate-buffered saline solution and flash frozen in liquid nitrogen. Membrane proteins were solubilized in 2% lauryl maltose neopentyl glycol (LMNG, Anatrace), 0.2% cholesteryl hemisuccinate tris salt (CHS, Anatrace), 20 mM HEPES pH 7.5, 150 mM KCl supplemented with protease-inhibitor cocktail (1 mM PMSF, 2.5 µg/mL aprotinin, 2.5 µg/mL leupeptin, 1 µg/mL pepstatin A) and spatula of DNaseI for 1 hr. Solubilized proteins were separated by centrifugation 75,000 $g$ for 40 min, followed by binding to 2.5 ml anti-GFP nanobody resin for 1 hr, which was equilibrated with washing buffer containing 0.01% LMNG, 50 mM Tris-HCl pH 8, 150 mM KCl, 2 mM DTT (BufferB). Anti-GFP nanobody affinity chromatography was performed by 20 column volumes of washing with BufferB, followed by overnight PreScission digestion, and elution with wash buffer. Protein sample was concentrated to a volume of 500 µl using CORNING SPIN-X concentrators (100 kDa cutoff), followed by centrifugation 10,000 g for 10 min. Concentrated proteins were further purified by size exclusion chromatography on a Superose 6 Increase 10/300 GL (GE healthcare) in BufferB. Peak fractions were pooled and concentrated to ~2.5 mg/mL using CORNING SPIN-X concentrators (100 kDa cutoff).

## Electron microscopy sample preparation and data acquisition

For CLC-7 from *Gallus gallus* (ggCLC-7), 3 µl of purified protein at a concentration of 2 mg/ml was applied to glow-discharged Au 400 mesh QUANTIFOIL R1.2/1.3 holey carbon grids (Quantifoil), and then plunged into liquid nitrogen-cooled liquid ethane with a FEI Vitrobot Mark IV (FEI Thermo Fisher). Grids were transferred to a 300 keV FEI Titan Krios microscopy equipped with a K2 summit direct electron detector (Gatan). Images were recorded with Leginon (*Suloway et al., 2005*) in super-resolution mode at 22,5000x, corresponding to pixel size of 0.536 Å. Dose rate was eight electrons/pixel/s, and defocus range was −1.2 to −2.5 µm. Images were recorded for 8 s with 0.2 s subframes (total 40 subframes), corresponding to a total dose of 61 electrons/Å$^2$.

For the CLC-7/OSTM1 complex from *Homo sapiens* (hsCLC-7/OSTM1), 3 µl of purified protein at a concentration of 2 mg/ml was supplemented with 1 mM ATP and 0.1% LMNG and was applied to glow-discharged Au 400 mesh QUANTIFOIL R1.2/1.3 holey carbon grids (Quantifoil), and then plunged into liquid nitrogen-cooled liquid ethane with a FEI Vitrobot Mark IV (FEI Thermo Fisher). Grids were transferred to a 300 keV FEI Titan Krios microscopy equipped with a K3 summit direct electron detector (Gatan). Images were recorded with SerialEM (*Mastronarde, 2005*) in super-resolution mode at 22,5000x, corresponding to pixel size of 0.532 Å. Dose rate was 13 electrons/pixel/s, and defocus range was −1.2 to −2.7 µm. Images were recorded for 4 s with 100 ms subframes (total 40 subframes), corresponding to a total dose of 44 electrons/Å$^2$.

## Electron microscopy data processing

40-frame super-resolution movies (0.536 Å/pixel) of ggCLC-7 were gain corrected, Fourier cropped by two and aligned using whole-frame and local motion correction algorithms by MotionCor2 (*Zheng et al., 2017*) (1.0723 Å/pixel). Whole-frame CTF parameters were determined using CTFfind 4.1.10 (*Rohou and Grigorieff, 2015*). Approximately 500 particles were manually selected to generate initial templates for autopicking that were improved by several rounds of two-dimensional

classification in Relion 3.0 (*Scheres, 2016*), resulting in 6,542,536 particles. False-positive selections and contaminants were excluded from the data using multiple rounds of heterogeneous classification in cryoSPARC v2 (*Punjani et al., 2017*) using models generated from the ab initio algorithm in cryoSPARC v2, resulting in a stack of 343,094 particles. Heterogeneous classification in cryoSPARC v2 was then used to identify 137,234 particles displaying both the transmembrane and cytosolic domains. After particle polishing in Relion and local CTF estimation and higher order aberration correction in cryoSPARC v2, a reconstruction was determined at resolution of 2.9 Å by non-uniform refinement in cryoSPARC v2 (*Punjani et al., 2019*). The final reconstruction was further improved by employing density modification on the two unfiltered half-maps with a soft mask in Phenix (*Terwilliger et al., 2019*).

40-frame super-resolution movies (0.532 Å/pixel) of hsCLC-7/OSTM1 complex were gain corrected, Fourier cropped by two and aligned using whole-frame and local motion correction algorithms by MotionCor2 (1.064 Å/pixel). Approximately 500 particles were manually selected to generate initial templates for autopicking that were improved by several rounds of two-dimensional classification in Relion and autopicking using Relion, resulting in 15,288,379 particles. False-positive selections and contaminants were excluded through iterative rounds of heterogeneous classification in cryoSPARC v2 using models generated from the ab initio algorithm in cryoSPARC v2, resulting in a stack of 932,232 particles. Heterogeneous classification in cryoSPARC v2 was then used to identify 327,619 particles displaying the luminal, transmembrane and cytosolic domains. After particle polishing in Relion and local CTF estimation and higher order aberration correction in cryoSPARC v2, a reconstruction was determined to 2.8 Å. 3D variability analysis in cryoSPARC v2 was then employed to characterize conformational heterogeneity (*Punjani and Fleet, 2020*). To interpret the results of the 3D variability analysis, CLC-7 and the luminal domain of OSTM1 were rigid-body docked into the two extreme states and the midpoint. Masks were generated for the luminal domain and the transmembrane and cytosolic domains that were used for local refinement. Local refinements yielded a reconstruction for the luminal domain at an estimated resolution of 3.0 Å, a reconstruction for the transmembrane domain at an estimated resolution of 2.9 Å and the cytosolic domain at an estimated resolution of 2.8 Å. The final reconstructions were then further improved by employing density modification on the two unfiltered half-maps with a soft mask in Phenix (*Terwilliger et al., 2019*). A composite map was generated from the local refinement maps in Phenix that was used for model building and refinement (*Terwilliger et al., 2019*).

## Model building and coordinate refinement

The structure of CLC from *Cyanidioschyzon merolae* (cmCLC) (*Feng et al., 2010*) was manually docked into the ggCLC-7 density map using chimera (*Pettersen et al., 2004*). The model was then manually rebuilt according to the density using coot (*Emsley et al., 2010*). Atomic coordinates were refined against the density modified map using phenix.real_space_refinement with geometric and Ramachandran restraints maintained throughout (*Adams et al., 2010*).

The refined ggCLC-7 structure was manually docked into the CLC-7/OSTM1 density map using Chimera (*Pettersen et al., 2004*). The human CLC-7 model was manually rebuilt using COOT to fit the density. OSTM1 was manually built into the density by first placing poly-alanine helices and then using large side chains and glycosylation sites to register the helices. N-linked glycosylation trees were built and refined using the 'carbohydrate' module in COOT (*Emsley and Crispin, 2018*). Notably, the density for the carbohydrate residues was of poorer quality than the nearby protein and due to the difficulty in modeling carbohydrates in cryo-EM density maps (*Emsley and Crispin, 2018*) they are less precisely modeled than the protein. Atomic coordinates were refined against the density modified map using phenix.real_space_refinement with geometric and Ramachandran restraints maintained throughout (*Adams et al., 2010*).

## Figures

Figures were prepared with UCSF Chimera (*Pettersen et al., 2004*), UCSF ChimeraX (*Goddard et al., 2018*), MOLE (*Pravda et al., 2018*), PyMol (*Schrödinger LLC, 2020*) and Jalview (*Waterhouse et al., 2009*).

## Acknowledgements

We thank M de la Cruz at the MSKCC Richard Rifkind Center and the Simons Electron Microscopy Center staff for cryo-EM for assistance with data collection, the MSKCC HPC group for assistance with data processing, members of the Hite lab for assistance with image acquisition and processing and A Accardi for comments on the manuscript. Some of this work was performed at the Simons Electron Microscopy Center and National Resource for Automated Molecular Microscopy located at the New York Structural Biology Center, supported by grants from the Simons Foundation (SF349247), NYSTAR, and the NIH National Institute of General Medical Sciences (GM103310) with additional support from the Agouron Institute (F00316) and NIH (OD019994). This work was supported in part by NIH-NCI Cancer Center Support Grant (P30 CA008748), the Josie Robertson Investigators Program (to RKH) and the Searle Scholars Program (to RKH).

## Additional information

### Funding

| Funder | Grant reference number | Author |
| --- | --- | --- |
| Searle Scholars Program | | Richard K Hite |
| Josie Robertson Investigators Program | | Richard K Hite |
| National Cancer Institute | accessory subunit | Richard K Hite |

The funders had no role in study design, data collection and interpretation, or the decision to submit the work for publication.

### Author contributions

Marina Schrecker, Conceptualization, Data curation, Formal analysis, Validation, Investigation, Visualization, Methodology, Writing - original draft, Writing - review and editing; Julia Korobenko, Data curation, Formal analysis, Investigation; Richard K Hite, Conceptualization, Formal analysis, Supervision, Funding acquisition, Visualization, Writing - original draft, Project administration, Writing - review and editing

### Author ORCIDs

Marina Schrecker https://orcid.org/0000-0001-8542-6657
Richard K Hite https://orcid.org/0000-0003-0496-0669

### Decision letter and Author response

Decision letter https://doi.org/10.7554/eLife.59555.sa1
Author response https://doi.org/10.7554/eLife.59555.sa2

## Additional files

### Supplementary files

• Transparent reporting form

### Data availability

Cryo-EM maps and atomic coordinates have been deposited with the EMDB and PDB under accession codes EMD-22386 and PDB ID 7JM6 for ggCLC-7 and EMD-22389 and PDB ID 7JM7 for human CLC-7/OSTM1.

The following datasets were generated:

| Author(s) | Year | Dataset title | Dataset URL | Database and Identifier |
| --- | --- | --- | --- | --- |
| Schrecker M, Hite | 2020 | Structure of chicken CLC-7 | https://www.rcsb.org/ | RCSB Protein Data |

| | | | | |
|---|---|---|---|---|
| RK | | | structure/7JM6 | Bank, 7JM6 |
| Schrecker M, Hite RK | 2020 | Structure of chicken CLC-7 | https://www.ebi.ac.uk/pdbe/entry/emdb/EMD-22386 | Electron Microscopy Data Bank, EMD-22386 |
| Schrecker M, Hite R | 2020 | Structure of human CLC-7/OSTM1 complex | https://www.rcsb.org/structure/7JM7 | RCSB Protein Data Bank, 7JM7 |
| Schrecker M, Hite R | 2020 | Structure of human CLC-7/OSTM1 complex | https://www.ebi.ac.uk/pdbe/entry/emdb/EMD-22389 | Electron Microscopy Data Bank, EMD-22389 |

The following previously published datasets were used:

| Author(s) | Year | Dataset title | Dataset URL | Database and Identifier |
|---|---|---|---|---|
| Dutzler R, Campbell EB, MacKinnon R | 2003 | Structure of the *Escherichia coli* ClC Chloride channel and Fab Complex | https://www.rcsb.org/structure/1OTS | RCSB Protein Data Bank, 1OTS |
| Dutzler R, Campbell EB, MacKinnon R | 2003 | Structure of the *Escherichia coli* ClC Chloride channel E148Q mutant and Fab Complex | https://www.rcsb.org/structure/1OTU | RCSB Protein Data Bank, 1OTU |
| Feng L, MacKinnon R | 2010 | Crystal Structure of a eukaryotic CLC transporter | https://www.rcsb.org/structure/3ORG | RCSB Protein Data Bank, 3ORG |
| Maduke M, Mathews II, Chavan TS | 2020 | Crystal structure of ClC-ec1 triple mutant (E113Q, E148Q, E203Q) | https://www.rcsb.org/structure/6V2J | RCSB Protein Data Bank, 6V2J |

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
