## [Decision Letter]

**Acceptance summary:**

The structures presented in this manuscript provide interesting new insights into this fascinating family of membrane transporters and channels. They show binding sites for ATP, which is known to regulate some CLCs, and for phosphatidyl inositol, which is not known to regulate CLCs but is of interest given the conserved binding site. In addition, the structures demonstrate how OSTM1 likely stabilizes CLC-7 in lysosomes and osteoclasts, where CLC-7 is expressed physiologically.

**Decision letter after peer review:**

Thank you for submitting your article "Cryo-EM structure of the lysosomal chloride-proton exchanger CLC-7 in complex with OSTM1" for consideration by *eLife*. Your article has been reviewed by two peer reviewers, and the evaluation has been overseen by a Reviewing Editor and Richard Aldrich as the Senior Editor. The following individual involved in review of your submission has agreed to reveal their identity: Raimund Dutzler (Reviewer #2).

The reviewers have discussed the reviews with one another and the Reviewing Editor has drafted this decision to help you prepare a revised submission.

Summary:

Schrecker et al. present cryo-EM structures of the CLC-7 chloride/proton antiporter. Chicken CLC-7 is solved without the accessory subunit OSTM1, and human CLC-7 is solved in complex with OSTM1. These are the first structures of a mammalian CLC transporter homolog and the first of any CLC with an accessory subunit. The data are of high quality. The structures reveal a mechanism by which OSTM1 stabilizes CLC-7 in lysosomes and osteoclasts (where CLC-7 is expressed physiologically, playing key roles in lysosomal pH regulation and osteoclast bone resorption). The cytoplasmic domain of ClC-7 carries a nonprotein density that the authors suggest, with strong support, represents a bound Mg-ATP. A highly interesting aspect of the new structure is another bound ligand, modeled as the phophoinositide PI3P. Although identification of PI3P in the sample by chemical methods or mass-spectrometry would further confirm its presence in the sample, the proposed binding model is reasonable given the shape of the residual density and the abundance of basic amino acids in the pocket; thus, additional support of this model is not essential at this time. The identification of this putative site for phosphatidyl inositol, which is not known to regulate CLCs, is of high interest given the conserved nature of the binding site. Altogether, these structures offer important new insights into CLC-7 molecular properties and provide a template for future structure-based functional investigations.

Essential revisions:

1) The moderate effects of OSTM1 binding to the human CLC-7 structure, as judged from the comparison with the chicken CLC-7 structure, is remarkable. The proposal that the OSTM interaction might alter the conformational properties of two Phe residues in vicinity of the ion binding path is interesting but it could also be a consequence of local differences between the two different orthologs. Moreover, the claim that this conformational difference might be connected to increased transport activity of the complex is noteworthy but speculative.

2) The authors point out the heterogeneity in orientation of the lumenal domain of OstM1, yet we see only one orientation in the final structure. It would be interesting for the reader to know something about the scale of this heterogeneity--is it tilting by a few angstroms or rocking strongly back and forth? Would it be possible to provide some kind of analysis to reveal something like the envelope of possible orientations? How might all this impact ion access to the transporter? Could there be electrostatic effects of OstM1?

3) The assignment of residual density in the data to bound chloride ions is justified in light of previously characterized ion binding sites of other CLC proteins. It is very interesting to see all three binding sites identified in the CLC from *E. coli* occupied in the structure of chicken CLC-7. It is also interesting to see that the density of the central Cl^-^ gets weaker in the structure of the human CLC-OSTM1 complex but it is unclear whether this reflects a lower occupancy or higher mobility of the ion or whether it is a consequence of the slightly lower quality of the data.

4) The interaction of ATP with the cytoplasmic domain of CLC-7 is exciting and it provides further evidence on the role of nucleotide binding to the cytoplasmic domains of certain CLC transporters. The interaction of the nucleotide with the N-terminus of the protein, which coordinates phosphate groups and Mg^2+^ is convincing and provides novel insight into the interaction, which was not revealed in the structure of the isolated domain of CLC-5. The assumption that ATP would bind with higher affinity to full-length CLC-7 than to the isolated cytoplasmic domain of CLC-5, is supported by the fact that the nucleotide is copurified with the sample. In light of the high intracellular ATP concentration, the proposal that ATP might form an integral part of the structure that remains bound instead of a regulatory ligand which dissociates from the protein is reasonable but would ultimately have to be characterized in functional experiments. In that respect it might be interesting to know whether a similar role of ATP as structural component has been found in other proteins.

5) The PI3P in the chicken structure is very interesting and novel, yet the authors say nothing about whether they did or did not observe a similar lipid in the human structure. Especially given their argument about the conservation of the lipid binding site, it is worth noting explicitly what they see here. In the same vein, and considering their argument about PI(3,5)P2, could there be differences between homologs in this site?

6) The non-protein density modeled as CHS is less convincing than the modeling of the PI3P site, and its model displayed in Figure 5—figure supplement 2 appears distorted.

---

## [Author Response]

Essential revisions:1) The moderate effects of OSTM1 binding to the human CLC-7 structure, as judged from the comparison with the chicken CLC-7 structure, is remarkable. The proposal that the OSTM interaction might alter the conformational properties of two Phe residues in vicinity of the ion binding path is interesting but it could also be a consequence of local differences between the two different orthologs. Moreover, the claim that this conformational difference might be connected to increased transport activity of the complex is noteworthy but speculative.

We thank the reviewer for the acknowledgement of their positive assessment of our work. We agree with the reviewers that it is possible that the subtle differences in the resolved conformations of the phenylanine side chains may be due to local differences between the human and chicken orthologs. We also agree with the reviewers that the connection between the conformation of the phenylanine residues and increased transport activity is highly speculative, and we have modified the manuscript in the subsection “Effects of OSTM1 binding to CLC-7” to note these points.

2) The authors point out the heterogeneity in orientation of the lumenal domain of OstM1, yet we see only one orientation in the final structure. It would be interesting for the reader to know something about the scale of this heterogeneity--is it tilting by a few angstroms or rocking strongly back and forth? Would it be possible to provide some kind of analysis to reveal something like the envelope of possible orientations? How might all this impact ion access to the transporter? Could there be electrostatic effects of OstM1?

We agree with the reviewer that it would be beneficial for the reader to include more details on the nature of OSTM1 movement. For a deeper understanding of the heterogeneity of the CLC-7/OSTM1 complex, we performed 3D variability analysis using CryoSPARC without imposing symmetry. One of the primary components of heterogeneity identified was a rotation of the luminal domain of OSTM1 with respect to CLC-7. We have added a new figure, Figure 5—figure supplement 2, which compares the two extremes states with the midpoint. To quantify the displacement of OSTM1 in the three states, we docked in and rigid-body fit the luminal domain of OSTM1 separately from CLC-7. When aligned by the transmembrane domains, the periphery of luminal domain is displaced by up to 6 Å when the two extreme states are compared. The displacement is sufficient to blur the high-resolution features of the density map, but is unlikely to obstruct access of ions to the conduction pathway and thus it is unclear if the flexibility has any functional consequences.

3) The assignment of residual density in the data to bound chloride ions is justified in light of previously characterized ion binding sites of other CLC proteins. It is very interesting to see all three binding sites identified in the CLC from *E. coli* occupied in the structure of chicken CLC-7. It is also interesting to see that the density of the central Cl^-^ gets weaker in the structure of the human CLC-OSTM1 complex but it is unclear whether this reflects a lower occupancy or higher mobility of the ion or whether it is a consequence of the slightly lower quality of the data.

We agree with the reviewer that our ability to interpret the densities that we attribute to the Cl^-^ ions CLC-7/OSTM1 density map is hindered by its anisotropy and the resulting lower quality. Thus, we can only speculate that the lower density of the central Cl^-^ ion is a result of lower occupancy or higher mobility. We have revised the text to “However, unlike in ggCLC-7, the central site in CLC-7/OSTM1 is the weakest and is only slightly above the background (~4 σ). While we must be cautious in interpreting the densities occupying the Cl^-^ binding sites of CLC-7/OSTM1 because of the anisotropy present in the data set, the differences in relative intensities of the Cl^-^ binding site peaks between ggCLC-7 and CLC-7/OSTM1 suggest that there may be a change in Cl^-^ occupancy of the central site when CLC-7 is bound to OSTM1. A change in occupancy of the central Cl^-^ site may be associated…”

4) The interaction of ATP with the cytoplasmic domain of CLC-7 is exciting and it provides further evidence on the role of nucleotide binding to the cytoplasmic domains of certain CLC transporters. The interaction of the nucleotide with the N-terminus of the protein, which coordinates phosphate groups and Mg^2+^ is convincing and provides novel insight into the interaction, which was not revealed in the structure of the isolated domain of CLC-5. The assumption that ATP would bind with higher affinity to full-length CLC-7 than to the isolated cytoplasmic domain of CLC-5, is supported by the fact that the nucleotide is copurified with the sample. In light of the high intracellular ATP concentration, the proposal that ATP might form an integral part of the structure that remains bound instead of a regulatory ligand which dissociates from the protein is reasonable but would ultimately have to be characterized in functional experiments. In that respect it might be interesting to know whether a similar role of ATP as structural component has been found in other proteins.

We appreciate the acknowledgement of the interest regarding the ATP-Mg^2+^-coordination of CLC-7. We absolutely agree that further studies are required for the ultimate characterization of CLC-7 regulation by ATP/Mg^2+^. Whole-cell recordings by Leisle et al., 2011 showed that CLC-7/OSTM1 remains fully active in the absence of ATP, however due to the predicted slow off-rate, endogenous ATP likely remained bound in their recordings.

We have included in the Discussion several examples of proteins that are proposed to bind nucleotides as structural components including AMPK, IP_3_ receptors and some of the prokaryotic RCK domain containing channels.

5) The PI3P in the chicken structure is very interesting and novel, yet the authors say nothing about whether they did or did not observe a similar lipid in the human structure. Especially given their argument about the conservation of the lipid binding site, it is worth noting explicitly what they see here. In the same vein, and considering their argument about PI(3,5)P2, could there be differences between homologs in this site?

We thank the reviewers for pointing out that we were unclear in our presentation of the ligands in the CLC-7/OSTM1 complex. To more clearly discuss the PI3P binding site within the CLC-7/OSTM1 complex. We have introduced a new figure (Figure 5—figure supplement 4) depicting an overlay of the ATP and lipid binding pockets of CLC-7 and CLC-7/OSTM1. We also show the density for the lipids in Figure 5—figure supplement 3. In the text we now state “Moreover, densities corresponding to ATP and PI3P molecules were resolved in their respective binding sites and the ligands interact with CLC-7 in a similar fashion regardless of the presence or absence of OSTM1 (Figure 5—figure supplements 3 and 4)” to emphasize that the ligand binding sites are conserved.

6) The non-protein density modeled as CHS is less convincing than the modeling of the PI3P site, and its model displayed in Figure 5—figure supplement 2 appears distorted.

We agree with the reviewer that modeling of CHS into the density is speculative. To avoid introducing potential errors into the model, we have removed the CHS molecule and have revised the text to “In addition to the direct protein-protein interactions, a non-protein density that may correspond to a cholesterol based on its size and shape was resolved at the interface between transmembrane domains of CLC-7 and OSTM1 (Figure 7B). Notably, no density was present at this site in the ggCLC-7 map, suggesting that CLC-7 and OSTM1 together may form an additional lipid binding site.”. We have depicted the position of the density in the complex in a new figure (Figure 5—figure supplement 3C).